

# Integrating multiple satellite observations into a coherent dataset to monitor the full water cycle - Application to the Mediterranean region

Victor Pellet[1,2], Filipe Aires[1,2], Simon Munier[3], Gabriel Jordá[4], Diego Fernández Prieto[5], Wouter Arnoud Dorigo[6], Jan Polcher[7], and Luca Brocca[8]

[1]LERMA, Observatoire de Paris, Paris, France.
[2]Estellus, Paris, France.
[3]CNRM, Météo-France, Toulouse, France.
[4]University of the Balearic Islands, Majorque, Spain.
[5]European Space Agency (ESA-ESRIN), Rome, Italy.
[6]Department of Geodesy and Geoinformation, TUWIEN, Wien, Austria.
[7]Laboratoire de Météorologie Dynamique, Paris, France.
[8]Istituto di ricerca per la protezione idrogeologica, Perugia, Italy.

**Abstract.** Integration techniques are used to combine Earth Observation (EO) datasets to study the Water Cycle (WC). By merging several datasets, they reduce uncertainty and introduce coherency among them. Several EO integration methods are presented and compared: The Optimal Selection (OS) simply choses the best individual datasets. Simple Weighting (SW) is a weighted sum of the datasets to reduce uncertainties. Three other techniques introduce a closure-constraint on the WC budget:

(1) The SW plus Post-Filtering (PF) is very efficient but it is applied at the basin-scale only, and lacks in spatial information. (2) By using a spatial interpolation scheme, the INTegration (INT) solution allows obtaining a pixel-scale database, but only for the common period of the all the water components. (3) A simple CALibration (CAL) of the EO datasets is therefore introduced to reproduce the INT results over the longer temporal extent of the EO datasets, but its closure constraint is relaxed. Results are presented over the Mediterranean region, one of the more complex environnements and a hot-spot for climate

change. We extended previous techniques to close simultaneously the terrestrial, oceanic and atmospheric WC budgets. We also introduce temporal and spatial multi-scaling constraints. The evaluation is performed for precipitation and evapotranspiration: in addition to better close the WC budget, the integrated database is also closer to *in situ* measurements. The resulting integrated database provides new estimates for the WC components: stock and flux annual-means are re-evaluated, and we now estimate the Bosporus net-flow mean value at 129 mm.yr$^{-1}$ for the 2004-2009 period. This new EO-based database describing the

terrestrial, oceanic and atmospheric WC over the Mediterranean is now proposed to the scientific community.

## 1 Introduction

The Mediterranean region is one of the main climate change hotspots (IPCC, 2014): its sensitivity to global change is high and its evolution remains uncertain. Its role in the evolution of the global ocean (i.e. mainly salinization and warming), as well as the socio-economics consequences it has for surrounding countries, stress the need of monitoring its water ressource. Analysing





the water cycle, the exchange among its terestrial, atmospheric and oceanic branches are critical to estimate the availability of the water in the Mediterranean region. A state-of-the-art in our understanding and quantification of Mediterranean water budget can be found for instance in Jordà et al. (2017). There, the various components are estimated using mainly reanalysis and regional climate modeling. The authors acknowledge that the range of uncertainty considering the water budget remains

high. In particular, the uncertainty for particular terms such as the direct estimate of the Strait of Gibraltar water transport is still a challenge. They recommend an increase of the observation of the air-Sea fluxes and a coordinated effort to gather numerical models outputs and observational discrete data. For the direct observation of the WC, the Mediterranean region is an uneven place. If the Europe presents a large network for *in situ* measurements of some water components such as precipitation and water vapour, the African coast are a data-sparse region. In this context, EO present the unique opportunity to observe at large

spatial and fine temporal scale the full picture of the WC in this region.

The WC can be described by the following time-varying budget equations:

$$
\begin{aligned}
\frac{\delta S_l}{\delta t} &= P_l - E_l - R_l \ \text{(Terrestrial)} \\
\frac{\delta S_o}{\delta t} &= P_o - E_o + R_l^* - Gib \ \text{(Oceanic)} \\
\frac{\delta W}{\delta t} &= E_{l/o} - P_{l/o} - Div \ \text{(Atmospheric)}
\end{aligned}
\tag{1}
$$

where $l$ stands for land and $o$ for ocean. $P$ is the precipitation; $E$ the evapo(transpi)ration; $S$ the water storage; $R_l$ the discharge from land to the Seas; $Gib$ is the net water transport at Gibraltar from the Mediterranean Sea into the Atlantic Ocean; $W$ the atmospheric water storage; and $Div$ the atmospheric vertically integrated moisture divergence. If all the components in Eq. (1) are expressed in mm/month (area-normalized) then a fourth equality is defined: $R_l^* = \frac{A_{land}}{A_{Sea}} \cdot R_l$ for total freshwater

input/output with $A_{land}$ is the total drainage area of the Mediterranean with the Black Sea, and $A_{Sea}$ the total of the two Sea areas.

In Pellet et al. (2017), satellite data are used to monitor the full water cycle over the Mediterranean region, but it is shown that the budget closure is not obtained with the EO datasets and that some integration technique should be used to optimize them. The use of EO data for WC monitoring remains a challenge for several reason: (1) the inherent uncertainties (system-

25 atic and random errors) of the satellite retrievals, (2) the multiplicity of datasets for the same geophysical parameter, (3) the inconsistency between datasets (for the same component or among the components of the WC). The optimal combination of EO datasets to study the water cycle has recently received a lot of attention. The features of some "combination" methods are synthesized in Table 1 for a better comparison of approaches considered in the following.

*The "Princeton" approach* - Pan and Wood (2006) presented first a work in which the authors aimed at closing the water balance using EO products. In this work, EO datasets such as precipitation was assimilated into a land surface model (the Variable Infiltration Capacity, VIC) using the combination of a Kalman filter and a closure constraint (see Table 1). The resulting "analysis" dataset is not a pure EO product since the VIC model is largely used. In fact, the authors show that the



Kalman filtering plus the closure constraint is equivalent to a traditional Kalman filtering, and then to the application of an independent post-filtering that constrains the closure (De Geeter et al., 1997; Simon and Tien Li Chia, 2002; Aires, 2014). This post-filtering acts by redistributing the budget residuals within each water component based on the uncertainties of each EO source. Several papers have been published based on this approach (Troy and Wood, 2009; McCabe et al., 2008; Sahoo et al.,

2011; Troy et al., 2011; Pan et al., 2012). For instance, in Sheffield et al. (2009), two different precipitation datasets were used over the Mississippi basin. Evapotranspiration was calculated using a revised Penman-Monteith formulation and changes in water storage were estimated from GRACE. For comparison, land surface model outputs, reanalyses data and *in situ* discharge measurements were used too. The authors concluded that a positive bias of the precipitation datasets leads to an overestimation of the discharge component when the estimation relies on EO data. Meanwhile, the land surface model shows a high degree

of agreement with *in situ* data. The analysis also highlights the importance of error characterization in the individual WC components. Yilmaz et al. (2011) relaxed the closure constraint during the assimilation. This is an important feature because strong closure constraint can result in high-frequency oscillations in the resulting combined dataset. A relaxed constraint is used in our approach (see Table 1).

*The NASA-NEWS initiative* - The project aims at a better characterization of the water cycle using EO data. The first step was

to improve the coherency of the satellite retrievals; then to gather the EO dataset, and calibrate them. Some information about the uncertainties of the EO datasets was gathered from the data producers, but these informations can not be straightforwardly used further in the integration process since their evaluation are not homogeneous but product-dependent. The water cycle budget can be closed using the satellite datasets (Rodell et al., 2015). However, this closure is obtained at the global and annual scale only, and residuals are still significant at regional and monthly scales (Rodell et al., 2015) uses then an interpolation for

a monthly closure. Closing the budget at the global scale was a first step,and closure must now be obtained at thinner spatial and temporal scales to monitor more precisely the distribution of the water components as the EO data are designed to. In Rodell et al. (2015), the storage terms (e.g. ground water storage) had no significant change when considering annual and global means. This hypothesis was then straightforwardly used at the monthly scale with an optimized interpolation scheme to relax the storage change at the monthly scale. This translates into a quadratic quality criterion where storage and fluxes terms

are minimized when using annual means, at the global scale (see Table 1). One interesting feature in this approach is that both the water (Rodell et al., 2015) and energy (L'Ecuyer et al., 2015) cycles were considered simultaneously in the assimilation taking into account the physical link between the two cycles through the Latent Heat flux.

*The ESA water cycle initiative* - In the context of the ESA WATCHFUL project on water budget closure, Aires (2014) developed several methodologies (Table 1) to integrate different hydrological datasets with a budget closure constraint, which

consisted in partitioning the budget residual, the imbalance of Eq.( 1), among the components, depending on the uncertainties of the respective datasets. No model was used in these integration methods, making the obtained product interesting for model calibration and validation. One of the proposed methods, the so-called Simple Weighting+Post Filtering (SW+PF), was then applied by Munier et al. (2014) over the Mississippi basin, using satellite datasets for $P$, $E$ and $\Delta S$ and gauge observations for $R$. The integrated components were compared to various *in situ* observations, showing good performances of the method.

The combination approach was performed at the entire basin level, but sub-basins could be considered if: river discharge is





available at the outlet, and the spatial domain is large enough for the spatial resolution of the GRACE measurements. One of the main limitations of the method relies on the datasets availability. Namely, datasets based on satellite observations generally cover the last two decades (except for GRACE measurements that started only in 2002), while discharge observations are not available over this time period for many large basins around the world. Hence, extending the method to the global scale, in

particular to un-gauged regions, is not straightforward. A Closure Correction Model (CCM) was then developed based on the integrated product (Munier et al., 2014), allowing to correct each dataset independently and to greatly reduce the budget residual. This calibration was applied over the basins where river discharges are available and extended to the global scale using an index characterizing the various surface types (Munier and Aires, 2017). This type of post-processing step is anchored in the combination approach, but it can be applied to long time records, at any time or spatial resolution. The result is an "anal-

ysis" of the various datasets. Some extensions of this work have been done and will be considered in this work. For instance, the integration is performed not only for the terrestrial water cycle, but also, simultaneously, for the atmospheric and oceanic components. Furthermore, the budget closure constraint is used simultaneously at different spatial (basin and sub-basin) and temporal (monthly and annual) scales.

Section 2 presents the study domain and introduces the datasets used in the following. The integration approaches are described together with the other combination techniques in Section 3. Section 4 presents the evaluation metrics for the integrated product: its ability to close the WC and its validation with *in situ* data at the sub-basin or pixel scale. Section 5 presents the water cycle analysis for the period 2004-2009 using our resulting integrated dataset. Finally, Section 6 concludes the analysis and presents some perspectives. All notations used in the following are summarized in Table A1 in appendix.

## 20  2   Case study and datasets

This section presents the spatial domain and the datasets used in this study. Table A1 in appendix summarizes the main characteristics of these products and more details can be found in Pellet et al. (2017). All products have different temporal extents but share a common coverage period 2004-2009.

### 2.1   Mediterranean region

The study domain is represented in Fig. 1. It is the catchment basin of the whole Mediterranean Sea drainage area, computed from each coastal pixel, including all rivers that flows into the Sea. Basins have been computed using a hydrographic model (Wu et al., 2011) at a spatial resolution of $0.25°$. The resolution of the hydrographic model used to compute land/Sea mask or catchment basin may have an impact on the spatial-average estimates and then on the WC budget residual. This area uncertainty is taken into account into the relaxation of the closure constrain at sub-basin scale (see Table 1). The Mediterranean Sea area

with the Black Sea is 3.0 million $km^2$, and its drainage area is more than 5 million $km^2$.

Sub-basins have been introduced in Pellet et al. (2017). They facilitate the analysis of local climate and specific hydrological features. The Mediterranean Sea and the terrestrial sub-basins used in the following are defined as:



- The west Maghreb mainly based on the Atlas mountain discharge (**MA-DZ-TN**);

- The Nile Basin and Libyan coast characterizing Saharan and sub-Saharan climate (**LY-EG**);

- The Spanish coasts and Pyrenees (**ES-Pyr**);

- The French coast, Italy and Adriatic Sea, freshwater from the Alps and the Balkans mountains (**Alp-IT-ADR**);

- The eastern part of the Mediterranean Sea, Greece, Turkey and Israel (**GR-TR-IL**);

- The whole Black Sea drainage catchment, Bulgaria, Georgia, Romania, Russia, Turkey, Ukraine, Slovakia, Hungary, Austria, Slovenia, Bosnia and Serbia (**BLS**).

In the current study, even if the closure methods (PF) is applied over the LY-EG sub-basin, the high uncertainty of the Nile discharge and its particular climate (African monsoon) as well as anthropogenic conditions (most of its water is used for
irrigation) make this sub-basin really different from the other sub-basins (Margat, 2004; Mariotti et al., 2002). In this study, the closure is ensured for the Nile sub-basin but no spatial extension will be extrapolated over the LY-EG and toward central Africa in the analysis (see Section 3).

## 2.2    Original EO datasets

The datasets presented in this section will be used in the integration process over the Mediterranean region. Most of them
are satellite products and are commonly used for studying the water cycle. In order to integrate them, the datasets have been projected on a common $0.25°$ spatial resolution grid, and re-sampled at the monthly scale.

*Precipitation* - Four satellite-based datasets have been selected. Two are gauge-calibrated products: the Tropical Rainfall Measuring Mission Multi-satellite Precipitation Analysis (TMPA, 3B42 V7) presented in Huffman et al. (2007) and the Global
Precipitation Climatology Project (GPCP, v2) introduced by Adler et al. (2003). Two are uncalibrated products: Joyce et al. (2004) has unveiled the NOAA CPC Morphing Technique (CMORPH, v1) and Ashouri et al. (2015) developed the Precipitation Estimation from Remote Sensing Information using Artificial Neural Network (PERSIANN, v1). In this study, we use a mix of gauged/ungauged-calibrated precipitation datasets. This choice is motivated by the will of preserving the original EO spatial pattern where limited gauge density in some areas may corrupt the signal during the gauge-calibration process (in
TMPA and GPCP product).

*Evapotranspiration* - Three satellite-based products were chosen to describe evapotranspiration (over land): the Global Land Evaporation Amsterdam Model (GLEAM-V3B, Martens et al., 2016; Miralles et al., 2011); the MODIS Global Evapotranspiration Project (MOD16, Mu et al., 2011); and the Numerical Terradynamic Simulation Group product (NSTG, Zhang et al.,
2010).

Two products were chosen for the evaporation (over the Sea): the Objectively Analyzed air-Sea Fluxes for Global Oceans




(OAflux, Sun et al., 2003); and The Global Energy and Water Cycle Exchanges Project product (GEWEX-Seaflux, Curry et al., 2004).

*Water storage change* - The terrestrial and Sea water storage datasets are all derived from the GRACE mission. The estimates of water storage implicitly include the underground water. Four satellite datasets are based on the spherical decomposition of GRACE measurement: the Jet Propulsion Laboratory (JPL, Watkins and Yuan, 2014) product; the Centre for Space ReSearch (CSR, Bettadpur, 2012) product, the German ReSearch Centre for Geoscience (GFZ, Dahle et al., 2013) product; and the land-only product from the Groupe de Recherche de Géodésie Spatiale (GRGS, Biancale et al., 2005). One extra solution based on the JPL-MASCONS decomposition of GRACE measurement (Watkins et al., 2015) is also used in this work. In order to compute the monthly change in water storage, we applied a centred derivative smoothing filter: [5/24 3/8 -3/8 -5/24] (Pellet et al., 2017). The filter is a slightly smoother version of the filter [1/8 1/4 -1/4 -1/8] presented by Eicker et al. (2016).Our filter has been compared with several other filters (results not shown). The chosen filter is a good compromise between its low smoothing (that suppress information) and its ability to de-noise the time serie.

*Discharge* - No satellite-based product is available for the discharge with sufficient temporal extent and few rivers are still monitored by public or private network for the Global Runoff Data Centre (GRDC) that collects discharge data at the global scale. The two discharge datasets used in the following are described in Pellet et al. (2017). Groundwater discharge is neglected and considered as uncertainty.

The CEFREM-V2 dataset of coastal annual discharge into the Mediterranean Sea (Ludwig et al., 2009) is based on *in situ* observations and some indirect estimates using the Pike formula (Pike, 1964). In addition, developed at the Laboratoire de Météorologie Dynamique (Polcher et al., 1998; Ducharne et al., 2003), the land surface model Organising Carbon and Hydrology In Dynamic Ecosystems (ORCHIDEE) is chosen here to describe the monthly dynamics of the discharge. Two coastal discharge outputs are available from its routing scheme with two different precipitation forcings: GPCC and Climatic ReSearch Unit (CRU) products. We therefore projected the monthly dynamical patterns from ORCHIDEE towards the CEFREM grid. We then scaled the monthly values of ORCHIDEE to match the CEFREM annual values. For comparison purpose, CEFREM total freshwater inflow into the Mediterranean, without the Black Sea is 400 Km$^3$ yr$^{-1}$; while ORCHIDEE is 380 Km$^3$ yr$^{-1}$. The scaling is then a simple way to take into account the anthropogenic impact that is not modelled at the annual and the 0.5° scales. The final product has then the spatial resolution and the annual cumulative value of CEFREM, but with the monthly dynamics of the ORCHIDEE model.

*Precipitable water change* - We considered two datasets for the precipitable water: the ESA Globvapor dataset (Schneider et al., 2013) and the 6-hour reanalysis product from the ECMWF reanalyses (ERA-I, Dee et al., 2011). The ERA-I reanalysis has been considered here because precipitable water, although model-based, is largely constrained by satellite observations. In order to compute changes in precipitable water, we also applied the derivative filter: [5/24 3/8 -3/8 -5/24].





*Moisture divergence* - Due to the limited temporal extent of the satellite-based data, we use the 6-hour ERA-I reanalysis product (Dee et al., 2011). Among the various re-analyses, ERA-I was adopted for this study in view of previous results demonstrating advantages in the representation of long term wind variability in Stopa and Cheung (2014) which plays a key role in the representation of moisture divergence. Nevertheless, Seager and Henderson (2013) have shown the limitation of the

reanalysis that do not catch moisture divergence events shorter than at the 6-hour temporal scale. This limitation must be taken into account when closing the water cycle.

*Gibraltar netflow* - The only multiannual estimate of the Gibraltar netflow based on observations is the one presented in Jordà et al. (2016). We use there a monthly reconstruction of the net transport where the effects of the atmospheric pressure have

been removed. This is done for consistency with the GRACE estimates of ocean water storage. The reconstruction technique used to generate that estimate has proven to be effective to simulate the variability but the uncertainties in the mean value are large. In Jordà et al. (2017) an expert-based assessment of the mean transport is presented. Therefore, in this work we substitute the 2004-2016 mean value of the Jordà et al. (2016) estimate by the estimate proposed in Jordà et al. (2017).
The Mediterranean Sea is also connected to the Red Sea with the Suez channel and to the Black Sea with the Strait of Bosporus.

The netflow at the Strait of Suez is neglected (Mariotti et al., 2002; Harzallah et al., 2016). Since no *in situ* reference is available on the Bosporus netflow, the current work gathers the Mediterranean and the Black Seas into a single reservoir.

## 2.3    Validation datasets

*The ENSEMBLES observation dataset (EOBS)* - In order to validate the precipitation, an additional dataset is used: the EOBS dataset developed from the EU-FP6 project ENSEMBLES (Haylock et al., 2008). It is a regional, well documented and vali-

dated *in situ* gridded daily dataset at $0.25°$ spatial resolution, covering the period 1950-2007.

*FLUXNET* - Ground-based FLUXNET data (Falge et al., 2017) are used to validate the evapotranspiration and precipitation over several sites in Europe [1]. These flux measurements are based on eddy covariance technique. All stations available in Europe for the 2004-2009 period have been selected. In order to avoid coastal contamination, the three Seaside towers "IT-Ro2",

"IT-Noe" and "ES-Amo" have been suppressed.

*Total and thermosteric Sea level datases*-To validate the Sea level output from the integration technique, we use and independent estimate of Mediterranean water content. The water content can be estimated (Fenoglio-Marc et al., 2006; Jordà and Gomis, 2006) as total Sea level minus the thermosteric variations (i.e. changes in Sea level due to thermal expan-

sion/contraction). Total Sea level is obtained from the Ssalto/Duacs altimeter data produced and distributed by the Copernicus Marine and Environment Monitoring Service [2] . The thermosteric Sea level variations are estimated using two ocean

---

[1]FLUXNET2015 datasets; https://fluxnet.fluxdata.org

[2]CMEMS http://www.marine.copernicus.eu





regional reanalyses (MEDRYS, Hamon et al., 2016; Bahurel et al., 2012, MyOcean,) and two global products that include the Mediterranean (the Met Office Hadley Centre EN-v4 Good et al., 2013; Ishii et al., 2003, ISHII).

## 2.4 EO uncertainty assumptions

Some studies aimed at characterizing the uncertainty of satellite retrieved products : estimating relative uncertainty of numerous datasets by the distance to the average product (Pan et al., 2012; Zhang et al., 2016) or using non-satellite datasets (Sahoo et al., 2011). Nevertheless, such characterizations are generally product- and site-specific, and for some products used in this work, no uncertainty characterization can be found in the literature. For these reasons we considered the same uncertainty for all the datasets of a given parameter after de-biasing, following Aires (2014).

Table 2 summarizes the uncertainty used in the various integration techniques. The uncertainty is associated to a weight which is the ratio of the sum of all the uncertainty in the WC equation and the uncertainty of the considered variable (computed as $\sigma_i^2 / \sum_i \sigma^2$ and expressed in percentage). Note that uncertainty in Table 2 stands for the merged product and not for particular satellite dataset (see Eq. (6)). Following Munier et al. (2014), the uncertainties are prescribed by the literature but slightly modified from Munier et al. (2014) to handle the special case of the Mediterranean region. Munier et al. (2014) used uncertainty values of 10 mm/month for each of the four $P$ products and the three $E$ products (leading to 5 and 5.8 mm/month for the merged P and E estimate), 5 mm/month for each of the three $\Delta S$ products (leading to 2.9 mm/month for the merged product) and 1 mm/month for only one $R$. The choice of these values was motivated by results of the studies cited in Section 1. In order to be closer to Rodell et al. (2015), on the one hand, we decide to reduce $P$ uncertainty to 4 mm/month. This is justified since the de-biasing was done toward the gauge-calibrated TMPA dataset (see Pellet et al. (2017) for details). On the other hand, we increased E uncertainty up to 6 mm/month. The uncertainty of the merged $\Delta S$ is estimate to be broadly the same since it is mainly driven by the large pixel resolution of GRACE. Finally, the uncertainty of the discharge $R$ has been increased since the product is partially based on model simulations and the groundwater discharge is not included in the analysis (see Section 2). For the atmospheric variables, we consider an uncertainty proportional to the range of variability for the precipitable water change: 1 mm/month. Following the suggestion from Seager and Henderson (2013), the reanalysis moisture divergence uncertainty has been set to 6 mm/month due to its large range of variability and time scale.

## 3 EO Merging methodologies

### 3.1 Closing the water cycle budget

In this section, the notations are introduced but additional details can be found in Aires (2014). We first consider the six terrestrial water components $X_l^t = (P_l, E_l, R_l, \Delta S_l, \Delta W_l, Div_l)$ and the six oceanic water components $X_o^t = (P_o, E_o, \Delta S_o, \Delta W_o, Div_o, Gib$ ($^t$ is the transpose symbol). We then define $X_{lo}^t = [X_l, X_o]^t$. The closure of the water budget can be relaxed using a centred



Gaussian random variable $r$ and $X^t \cdot G_{lo}^t = r$, $with\ r \sim \mathcal{N}(O, \sum)$ where:

$$G_{lo} = \begin{bmatrix} 1 & -1 & -1 & -1 & 0 & 0 & 0 & 0 & 0 & 0 & 0 & 0 \\ -1 & 1 & 0 & 0 & -1 & -1 & 0 & 0 & 0 & 0 & 0 & 0 \\ 0 & 0 & \frac{A_{land}}{A_{Sea}} & 0 & 0 & 0 & 1 & -1 & -1 & 0 & 0 & -1 \\ 0 & 0 & 0 & 0 & 0 & 0 & -1 & 1 & 0 & -1 & -1 & 0 \end{bmatrix} \qquad (2)$$

which is equivalent to the water budget in Eq. (1).

Let:

$$Y_l^t = \begin{matrix} (P_1, \ldots, P_p,\ E_1, \ldots, E_q,\ R_1, \ldots, R_r, \\ \Delta S_1, \ldots, \Delta S_s,\ \Delta W_1, \ldots, \Delta W_v,\ Div_1, \ldots, Div_d) \end{matrix} \qquad (3)$$

be the vector of dimension $n_l = p + q + m + s + v + d$ gathering the multiple observations available for each water component over land (similarly $Y_o$ of dimension $n_o$ is defined over Sea):

- $(P_1,\ P_2, \ldots,\ P_p)$, the $p$ precipitation estimates;

- $(E_1,\ E_2, \ldots,\ E_q)$, the $q$ sources of information for evapotranspiration;

- $(R_1,\ R_2, \ldots,\ R_m)$, the $m$ discharge estimates;

- $(\Delta S_1,\ \Delta S_2, \ldots,\ \Delta S_s)$, the $s$ sources of information for the water storage change;

- $(\Delta W_1,\ \Delta W_2, \ldots,\ \Delta W_v)$, the $v$ precipitable water change estimates;

- $(Div_1,\ Div_2, \ldots,\ Div_d)$, the $d$ moisture divergence.

The aim of this approach is to obtain a linear filter $K_{an}$ used to obtain an estimate $X_{an}$ ("$an$" stands for analysis) of $X_{lo}$ based on the observations $Y_{lo}$:

$$X_{an} = K_{an} \cdot Y_{lo}\ \ with\ \ Y_{lo} = [Y_l, Y_o] \qquad (4)$$

where $K_{an}$ is a $12 \times (n_l + n_o)$ matrix.

## 3.2 Optimal Selection (OS)

A first simple use of the EO datasets to study the WC is to determine the optimal selection of satellite datasets for each water component, based on the budget closure. This method has been used for instance in Pellet et al. (2017). The choice of the best combination of datasets (one dataset for eah water cycle component) is based on the lowest residual statistics for the WC budget. If Eq. (1) is not closed, there are residuals left, and a quality criterion intends to reduce them as much as possible. A systematic Search for the best combination is performed: for all possible combinations, the water residual is calculated over each sub-basin following Eq. (1). The combination of datasets with the lowest budget residual is then selected. This OS solution is the best use of the EO dataset without any merging technique.





### 3.3 Simple Weighing (SW)

The SW approach relies on the merging of the EO datasets for each water component contrarily to the OS method. EO products and more generally any estimation of a variable via observations, presents two types of errors. (1) Systematic errors related, for instance, to the absolute calibration of the sensor. These can be represented by a bias and/or a scaling factor. (2) Random

errors related to retrieval algorithm uncertainties or to missing or inaccurate auxiliary information (e.g cloud mask) or to the sensor itself. These are often characterized by a standard deviation using a Gaussian hypothesis. From a statistical point-of-view, using the average of several estimates reduces the random errors of the estimation if no bias errors are present in the estimates. The merging process such as in Eq. (4) requires then un-biased estimates (Aires, 2014). The difficulty is that, as for uncertainties (Section 2.2.4), it is rather difficult to obtain bias estimates from the literature for every dataset used in this

approach. A pragmatic strategy is to set the reference as the mean state for each component. Then, all the sources of information for this component are bias-corrected toward this reference (Munier and Aires, 2017). A slightly modified version of the bias correction is to choose one reference among the datasets and apply the bias-correction. The author opted for the modified version and de-biased the EO using the TMPA Season (Pellet et al., 2017). Therefore, the SW methodology, presented for instance in Aires (2014), is first based on a Seasonal bias correction to reduce the systematic biases and is then followed by a

weighted average of the corrected estimates to reduce the random errors.

The SW methodology uses the diversity of WC component estimations to reduce the random errors. Let us consider the $p$ precipitation observations $P_i$ associated with errors $\epsilon_i$ following a Gaussian distribution N(0, $\sigma_i$). The $\sigma_i$ is the standard deviation of the estimate $i^{th}$. The SW precipitation estimate $P_{SW}$ is given by the weighted average:

$$20 \quad P_{SW} = \frac{1}{p-1} \sum_{i=1}^{p} \frac{\sum_{k \neq i} (\sigma_k)^2}{\sum_k (\sigma_k)^2} P_i. \tag{5}$$

This equation is valid when there is no bias error in the $P_i$s (thanks to the preliminary bias correction) and is optimal when the errors $\epsilon_i$ are statistically independent from each other. This expression is valid for the other WC components. The variance of the $P_{SW}$ estimate is then given by:

$$var(P_{SW}) = \frac{1}{(p-1)^2} \sum_{i=1}^{p} \left( \frac{\sum_{k \neq i} (\sigma_k)^2}{\sum_k (\sigma_k)^2} \right)^2 \sigma_i^2. \tag{6}$$

This is an important information because it gives the uncertainty of the estimate of Eq. (5). It shows that the $P_{SW}$ errors can be significantly reduced by increasing the number $p$ of observations.

Following Eq. (5) the state vector estimate via SW method $X_{SW}$ can be defined as:

$$X_{SW} = K_{SW} \cdot Y_{lo}, \tag{7}$$

where $K_{SW}$ is a $12 \times (n_l + n_o)$ matrix in which each line represent Eq. (5) for one of the 12 water components (the first one

for the precipitation estimate, the second for the evapotranspiration, *ect.*) and based on the $(n_l + n_o)$ observations.





## 3.4 Post-Filtering (PF)

In the SW approach, each water component is weighted (see Eqs. (6-7)) based on its *a priori* uncertainties (Section 2) but no closure constraint is imposed on the solution $X_{SW}$. Several methods were considered in Aires (2014) to introduce a WC budget closure constraint on the SW solution. However, Monte-Carlo simulations have shown that the SW solution associated

to a so-called Post-Filtering (PF) provides results as good as more complex techniques such as variational assimilation.

The PF approach has been introduced (Pan and Wood, 2006) to impose the closure constraint on a previously obtained solution. Here we used $X_{SW}$ as the first guess on the state vector $X_{lo}$. In Aires (2014), the PF was used and tested without any model, as a simple post-processing step after the SW. Following Yilmaz et al. (2011), the current study implements the PF filter with a relaxed closure constraint characterized by its uncertainty covariance $\sum$:

$$X_{PF} = (I - K_{PF} \cdot G_{lo} \sum\nolimits^{-1} G_{lo}^t) \cdot X_{SW}, \tag{8}$$

where $K_{PF} = (B_{lo}^{-1} + G_{lo} \sum\nolimits^{-1} G_{lo}^t)^{-1}$ and $B_{lo}$ is the error covariance matrix of the first estimate on $X_{lo}$.

In expressing $X_{SW}$ with $Y_{lo}$, we can express explicitly the linear operator $K_{an}$ of Eq. (4):

$$X_{an} = X_{PF} = (I - K_{PF} \cdot G_{lo} \sum\nolimits^{-1} G_{lo}^t) \cdot K_{SW} \cdot Y_{lo},$$
$$X_{an} = K_{an} \cdot Y_{lo}, \tag{9}$$

Where $K_{an} = (I - K_{PF} \cdot G_{lo} \sum\nolimits^{-1} G_{lo}^t) \cdot K_{SW}$. The PF step (budget closure) consists in partitioning the budget residual among the twelve components at each time step, independently. This technique allows obtaining a satisfactory WC budget closure for each basin. The residual term $r$ could be reduced in SW+PF approach by decreasing the variance $\sum$ in Eq. (8). If the relaxation term is to small, the closure is constrained but this is to the detriment of some hypotheses (such as unknown ground water) and some uncertainties (e.g.size of the drainage area used to compute spatial average of the water components).

Following (Aires, 2014; Munier et al., 2014) we enforced the budget closure by frequency range to avoid high-frequency errors to impact the low-frequency variables such as evapotranspiration (mainly driven by annual vegetation growth (Allen et al., 1998)). We first decomposed each parameter into a high and low-frequency components considering a cut-off frequency of 6 months (using a FFT decomposition). The budget is then applied independently on low and high frequencies. The high frequency component of $E$ is then not included in the high budget closure. The linearity of PF and FFT ensures the budget

closure of the re-composed final product. In the following temporal multi-scaling, the annual constraint is applied only on the low-frequency budget closure.

*Spatial multi-scaling* - It is possible to impose a WC budget closure simultaneously over the six sub-basins, the full basin and over the ocean (i.e. Mediterranean and Black Seas). Let us consider the total WC state vector:

$$X^t = [X_l^{(1)}, \ X_l^{(2)}, \ X_l^{(3)}, \ X_l^{(4)}, \ X_l^{(5)}, \ X_l^{(6)}, \ X_o]^t. \tag{10}$$



that includes the six water components $X_l^i$ over each sub-basin $i$ of area $A_l^{(i)}$ and ocean. The "closure" matrix becomes:

$$G_{lo} = \begin{pmatrix} G_l^{(1)} & 0 & \cdots & 0 & 0 \\ 0 & G_l^{(2)} & \cdots & 0 & 0 \\ \cdots & \cdots & \cdots & \cdots & \cdots \\ 0 & 0 & \cdots & G_l^{(6)} & 0 \\ L_{lo}^{(1)} & L_{lo}^{(2)} & \cdots & L_{lo}^{(6)} & G_o \end{pmatrix} \tag{11}$$

with:

$$G_l^{(i)} = \begin{bmatrix} 1 & -1 & -1 & -1 & 0 & 0 \\ -1 & 1 & 0 & 0 & -1 & -1 \end{bmatrix}$$

$$L_{lo}^{(i)} = \begin{bmatrix} 0 & 0 & \frac{A_l^{(i)}}{A_{Sea}} & 0 & 0 & 0 \\ 0 & 0 & 0 & 0 & 0 & 0 \end{bmatrix} \tag{12}$$

$$G_o = \begin{bmatrix} 1 & -1 & -1 & 0 & 0 & 1 \\ -1 & 1 & 0 & -1 & -1 & 0 \end{bmatrix}$$

The last row of $G_{lo}$ represents the overall budget closure, including all the sub-basins and the ocean. The dimension of the covariance matrices $B_{lo}$ and $\sum$ are increased following the new size of the state vector $X_{lo}$. No cross terms in $B_{lo}$ and $\sum$ are included, meaning that there is no dependency of the first guess and closure errors among the sub-basins.

*Temporal multi-scaling* - It is also possible to impose a WC budget closure simultaneously at monthly and annual scales. With monthly closure, the annual closure should automatically be obtained but due to the relaxation of the closure constrain, the annual closure would be relawed too. We control here the yearly closure constrain with an uncertainty of 1 mm. Furthermore, we impose a yearly closure assuming no groundwater storage change at the annual scale over land (representing an additional constraint on $\Delta S_l$ to ensure that no bias is introduced for this variable during the PF process). In this framework, monthly closures are now interdependent in the given year and the new state vector is :

$$X_{year}^t = [X^{Jan}, \cdots, X^{Dec}]^t, \tag{13}$$

with $X^m$ is the total state vector X defined in Eq. (10), for month m. The closure is applied independently for the four years of the 2004-2009 period but the twelve months of each year are closed independently.

The closure matrix $GA_{lo}$ that includes closure for the twelve months of the year and the full year is derived from the monthly constraint Eq. 11 and defined as:




$$GA_{lo} = \begin{pmatrix} G_{lo} & 0 & \cdots & 0 \\ 0 & G_{lo} & \cdots & 0 \\ \cdots & \cdots & \cdots & \cdots \\ 0 & 0 & \cdots & G_{lo} \\ N_{lo} & N_{lo} & \cdots & N_{lo} \end{pmatrix} \tag{14}$$

where $N_{lo}$ is the modified closure matrix $G_{lo}$ in which the matrix $G_l^{(i)}$ is rewriten in $N_l^{(i)}$ by imposing $\Delta S_l = 0$ :

$$N_l^{(i)} = \begin{bmatrix} 1 & -1 & -1 & \mathbf{0} & 0 & 0 \\ -1 & 1 & 0 & 0 & -1 & -1 \end{bmatrix} \tag{15}$$

The last row of $GA_{lo}$ represents the annual budget closure considering no storage change at the annual scale over land, including all the sub-basins and the ocean. The dimension of the covariance matrices $B_{lo}$ and $\sum$ are increased once again following the new size of the state vector $X_{year}$. No cross terms in $B_{lo}$ and $\sum$ are included, meaning that there is no dependency of the first guess and closure errors between the months.

This SW+PF technique is able to deal only with time series (the average on the considered sub-basins), not with maps (pixel) since the discharge is not available at this resolution. Therefore, in order to obtain a multi-component dataset that closes the WC budget and has spatial patterns at the pixel level, another technique needs to be used.

### 3.5    INTegration (INT)

The INT methodology allows extrapolating the results obtained with the previous SW+PF, from the sub-basin to the pixel
scale. To obtain a pixel-wise closure, Zhang et al. (2017) assimilate satellite data into the VIC model at the pixel scale (0.5°) using the VIC pixel water storage and runoff information. Munier and Aires (2017) extrapolated at the global scale the results of the WC closure of several large river basins around the globe, by using surface classes that intend to discriminate between EO error types, preserving as much as possible the hydrological coherency.

   The INT approach proposed here uses the WC closure over the Mediterranean sub-basins to extrapolate the closure correction
to the surrounding area. The methodology is presented in its various steps in Fig. 2 for precipitation and evaporation, for a particular month. In this analysis, we consider only the Mediterranean sub-basins and their close neighbourhood, so a simple spatial interpolation of the closure correction is supposed to be sufficient.

   The SW+PF method (Fig. 2, second row) provides a WC budget closure over the six sub-basins, for each month $m = 1, \cdots, 72$ of the 2004-2009 period.

The INT method requires a scaling factor to go from the SW to the SW+PF solution at the sub-basin scale. We define $\beta^{(i)}(m) = P_{PF}^{(i)}(m)/P_{SW}^{(i)}(m)$ (for precipitation here), the ratio between the SW and the SW+PF solution, for each sub-basin



$i$ and month $m$. This ratio can be used to scale the SW dataset towards the SW+PF solution at the basin scale, for a particular month $m$, in the following way:

$$P_{INT}^{(i)}(m) = \beta^{(i)}(m) \cdot P_{SW}^{(i)}(m) \left( = P_{PF}^{(i)}(m) \right). \tag{16}$$

For water storage change or moisture divergence, this $\beta$ could become negative. In this case, the bias-correction $\gamma^{(i)}(m) = P_{PF}^{(i)}(m) - P_{SW}^{(i)}(m)$ is used instead:

$$\Delta S_{INT}^{(i)}(m) = \Delta S_{SW}^{(i)}(m) + \gamma^{(i)}(m) \left( = \Delta S_{PF}^{(i)}(m) \right). \tag{17}$$

The $\beta$ scaling is defined at the sub-basin scale, but if interpolated spatially, it could be used at the pixel scale to obtain a truly spatialized solution.

Let us define a scaling map at the pixel level $\alpha$ such that: for each pixel $j$ in sub-basin $i$, for each month $m$: $\alpha(j,m) = \beta^{(i)}(m)$ (or $\gamma^{(i)}(m)$). When used as it is, the convolution of SW and $\alpha$ maps allows for the spatialisation of the sub-basins closure (Fig. 2, third row) with :

$$\iint\limits_{j \in A_l^{(i)}} P_{SW}(j,m) \dot{\alpha}(j,m) = \beta^{(i)}(m) \cdot P_{SW}^{(i)}(m) = P_{INT}^{(i)}(m) \tag{18}$$

However, this product presents not only a discontinuity across the sub-basins (where different scaling factors $\beta$ are defined) but also no value can be provided outside of the sub-basins.

To solve these two issues, the $\alpha$ scaling maps are interpolated/extrapolated:

- *Interpolation* - A region of 200 km on either side of the frontier between two sub-basins $i_1$ and $i_2$ is defined, and a smooth interpolation is performed between the two scaling factors $\beta^{(i_1)}(m)$ and $\beta^{(i_2)}(m)$ based on the distance to the frontier. This interpolation of the scaling factors $\alpha$ between two sub-basins can introduce errors (closure residuals can slightly increase) but it will be shown that this effect is limited and that the bottom equations (in parenthesis) in Eqs. (16-17) stand overall.

- *Extrapolation* - An extrapolation of the $\alpha$ maps is then performed to have a scaling factor $\alpha$ outside of the sub-basins domain. This extrapolation is weighted according to the respective distances to the two closest sub-basins.

The INT product is the convolution between the SW EO dataset with the resulting scaling map $\alpha$ that constrains the WC budget closure. INT is then an optimised version of SW in which the WC budget closure has been extended at the pixel scale. The fourth row of Fig. 2 shows the resulting INT product and its spatial coverage. The continuity issues between the sub-basins have been solved, and the extrapolation allows for a spatial coverage over the entire domain. The difference between the SW and INT estimates, represented in the last row of Fig. 2, is then directly related to the pixel-wise interpolated scaling factor $\alpha$. Discontinuity between the sub-basins is smoothed. The north of Europe excluding France is mainly driven by the scaling factor on the BLS region. That is consistent with the updated köppen climate classification (Kottek et al., 2006). Since the SW+PF solution is available over the 2004-2009 period only, INT can be obtained only over this period.





## 3.6 CALibration (CAL)

To obtain the INT solution, many EO datasets were combined: multiple datasets for each water component (the SW part), and for the various WC components (the PF part). However, if one of the datasets is missing, the INT solution cannot be estimated and this will result in a gap in the time record, and shorter time series of the integrated database.

In Munier et al. (2014), a "Closure Correction Model" (CCM) was introduced to correct each dataset independently, based on the results of the SW+PF integration. The CCM is defined as a simple affine transformation with a scaling factor $a$ and an offset $b$, such that $X_{new} = a \cdot X + b$. The CCM parameters $a$ and $b$ were calibrated by computing a linear regression between the original observation datasets and the SW+PF components.

A similar approach can be used, with the INT solution as a reference instead of the SW+PF. Instead of calibrating the original EO datasets using basin scale data, we propose here to calibrate the SW solution towards the INT solution at the pixel scale. This calibration of the SW allows obtaining a long-term dataset at the pixel scale like the SW solution, see Table 3, but with WC budget closure statistics closer to the INT solution. In our tests (not shown), the linear regression is quite satisfactory for the calibration, and it is not necessary to use a more complex statistical regression tool such as a neural network.

The merging/integration techniques used in this study are described in Table 3.

## 4 Evaluation of the integrated datasets

### 4.1 Water cycle budget closure

The impact of hydrological constraint (PF) as well as the INTegration (INT) and CALibration (CAL) processes on the spatial averaging of the water component estimates and the WC budget residuals, over the several Mediterranean sub-basins, is summarized on Fig A1 in the Apendix.

The residuals of the surface and atmospheric WC budgets for the Mediterranean region are computed at the monthly scale, over the 2004-2009 period. The Root Mean Square (RMS) statistics of these residuals are summarized in Table 4 for the six considered products (ERA-I, OS, SW, SW+PF, INT and CAL). Percentage of improvement of the RMS of the residuals with respect to the SW solution are also shown for comparison purposes.

ERA-I stands for the reanalyses product for all variables except for the water storage and the discharge, to keep the comparison consistent. It should be noted that ERA-I does not have any water conservation constrain. The optimal selection is given by: TMPA precipitation; GLEAM evapotranspiration and OAFlux evaporation; GRGS water storage change over land and JPL water storage change over Sea; GPCC-forced ORCHIDEE-CEFREM discharge; and the derivated Globvapor for atmospheric water vapour change. Only one dataset is available for the moisture divergence (Pellet et al., 2017). As shown in (Aires, 2014; Munier et al., 2014; Pellet et al., 2017), the SW merging procedure reduces the WC budget residuals at the sub-basin scale, by reducing the random errors of the EO data. The product outperforms the ERA-I reanalysis and the OS product. However,



the full closure is generally not satisfactory with this technique. The SW+PF procedure closes the water budget over all the sub-basins, and over the surface and in the atmosphere, with a RMS of the residual of about 4 mm/month. The surface budget residuals are drastically reduced: from 72% over the GR-TR-IL sub-basin and up to 94% for the Mediterranean Sea. This shows the necessity to use a WC budget closure constraint that links the six water components.

5    The INT product provides satisfactory budget closure results (from 61% to 94%), even if they are slightly degraded compared to the SW+PF (due to the interpolation process between sub-basins). Since no interpolation has been applied over the Mediterranean Sea, the statistic are equal to the SW+PF.

The CAL product improves less the WC budget residuals compared to INT. Nevertheless, the RMS of the residuals for these products are reduced over all sub-basins compared to SW solution.

Fig. A1 gives, in Appendix, the 2004-2009 time series of all the water components estimate for the various methodologies (SW, SW+PF, INT and CAL) over the various sub-basin as well as the probability density function of the residual. This figure shows how the WC closure impact the time series.

## 4.2   Evaluation at the sub-basin scale

Since the WC budget closure constraint was imposed at the sub-basin scale (see Section 3), the evaluation of the integrated product is done at this scale too. Two metrics are used here, the RMS of the Difference (RMSD) with *in situ* measurements and the CORRelation (CORR). Only multiple-EO integrated datasets are compared in the two following sections.

*Terrestrial precipitation* - Table 5 provides the comparison of the EOBS gridded gauge precipitation dataset (section 2.2) with
the SW, SW+PF, INT and CAL solutions, in terms of temporal correlation (at the monthly and sub-basins scales), and RMSD, for each sub-basin and for the "continental" scale (land included in Fig. 1). Since the SW+PF product is defined only on the Mediterranean drainage sub-basins, no statistic is shown for this approach over the continental region (last column). For the RMSD error statistics, results are also provided as improvements compared to the SW solution.

Over all the sub-basins, the SW+PF methodology improves results compared to the un-constrained SW method. Even if
the correlation of SW with EOBS is already good, the closure constraint improves this correlation to 0.84 (0.81) over e.g. the MA-DZ-TN sub-basin. This is true even over the complex sub-basins Alp-IT-ADR. SW+PF also reduces the RMSD with EOBS (by up to 20%). These results show the merit of the closure constraint on precipitation. Without explicitly constraining satellite precipitation products towards the *in situ* data, SW+PF statistics are still improved.

The INT product shows similar CORR and RMSD statistics as SW+PF over the Mediterranean sub-basins, with a slight
decrease of the CORR with EOBS over the ES-Pyr sub-basin. Over the continental region, INT improves the correlation compared to SW (from 0.78 to 0.80) while reducing by 17% the RMSD. Therefore, the interpolation process between the sub-basins (see the spatialization in Section 3.3.5) does not degrade the solution inside the sub-basins, while the extrapolation outside of them allows to improve the un-constrained SW statistics over the whole continent. This is a true benefit since INT




presents comparable performances to SW+PF in terms of closure capability and closeness to *in situ* measurements, with the advantage of the spatial variability at the pixel scale.

Finally, the CAL precipitation product shows results as good as SW (slightly better for the whole continental region) for the CORR, and smaller RMSD with EOBS. The CAL product does not close as well the WC budget as the INT solution, but it has

the advantage of being available over a longer time-record (1980-2012) compared to the 2004-2009 INT period.

*Sea Water Level change* - The Seawater storage (related to the Sea water level) change over the Mediterranean Sea (excluding the Black Sea) is tested using altimetry and thermal datasets over the 2004-2009 period. First, the thermal content estimates of the four datasets presented in Section 2.2 are merged into one single estimate. The merge thermal content estimate is then

subtracted from the AVISO Altimetry Sea water level. The monthly change is then computed using the same derivative filter as the one used for GRACE:[5/24 3/8 -3/8 -5/24].

Fig. 3 shows the altimetry estimate and the various methodologies estimates. Since the Mediterranean Sea is considered without the Black Sea for this evaluation, there is no SW+PF estimate (that added the Mediterranean Sea and the Black Sea). While the SW solution has a 0.52 CORR and a 12.2 mm/month RMSD with respect to Altimetry estimate, INT statistic are

0.58 for the CORR and 11.8 mm/month for the RMSD and CAL 0.56 for the CORR and 11.8 mm/month for the RMSD. Here again, the INT estimate outperforms the unconstrained SW methodology in both CORR and RMSD. CAL presents also better results than SW but the CORR with altimetry is slightly reduced compared to INT. Using the closure of the Mediterranean and Black Seas improves the water storage change estimates.

### 4.3   Evaluation at the pixel scale

The INT and CAL estimates are here evaluated at the pixel scale, for precipitation and evapotranspiration. Improvements of SW by INT and CAL are measured using *in situ* measurements of precipitation and evapotranspiration from the FLUXnet database, available over the Mediterranean region, for the 2004-2009 period (section 2.2).

Fig. 4 presents the scatter-plots of the RMSD between the SW estimate ($E_sw$) and INT or CAL ($E_{cor}$ for "corrected") datasets with the FLUXnet evapotranspiration data ($E_{FLUX}$), for each station. The 1:1 line is also shown in scatter-plots.

This line characterizes the (un)improvement due to the closure: each dots under the 1:1 line represents an improvement at the corresponding station from SW solution to INT or/and CAL. INT and CAL improve evapotranspiration estimates for more than 53% of the sites. The distribution of the differences in the encapsulated figure is slightly narrowed by the INT and CAL compared to the SW solution. Location of the station where the closure improves the RMSD with the flux measurement is shown in green if INT and CAL both improves the estimate, yellow when only CAL improves, and blue when only INT

improves. Red dots represent station where there are a degradation in both INT and CAL. These results must be taken with caution since the characteristic of the FLUXnet sites may not be representative of the full satellite domain considered in our paper.

Fig. 5 presents the scatter-plots of the RMSD between the SW estimate ($P_sw$) and INT or CAL ($P_{cor}$ for "corrected") datasets with the FLUXnet precipitation data ($P_{FLUX}$), for each station. Over most of the stations (82%), the INT and CAL solutions



improve precipitation estimate compared to SW. Location of improved sites are shown with the same color code as in Fig. 4. It can be seen in Fig. 5 that red dots are located mainly in mountainous or coastal region. These two type of landscape are really challenging for precipitation estimate due to snow precipitation on one side and coastal Sea/land contamination on the other.

## 5   A coherent multi-component dataset for the water cycle monitoring

### 5.1   A quasi-triangular balance

The mean fluxes of the Mediterranean water cycle and associated variability, over the 2004-2009 period are depicted in Fig. 6. The variability is computed as the standard deviation of the annual values over the period. These value have been computed over the respective terrestrial or oceanic sub-basins; considering all the drainage area in Europe (within Turkish) or in Africa (without considering the Nile river basin for which just its discharge is represented), Black Sea or Mediterranean Sea. The large font numbers are the estimates resulting from the INT methodology while the little font is for SW. The two values for the netflow estimate at Bosporus strait are estimated as the deficit term of the water budget equation, computed over the Mediterranean and Black Seas independently. Using INT estimate (i.e. closure of the two Seas at once) the two values are in better agreement to each other than to the two SW estimate. In the following, only the INT values are described.

Fig. 6 shows the uneven water contribution between the European ($316\pm57$ km$^3$ yr$^{-1}$ for the total discharge) and the African ($83\pm30$ km$^3$ yr$^{-1}$ within the Nile discharge) drainage area to the Mediterranean Sea budget. Furthermore, it shows the role of the Black Sea in the global Mediterranean WC. Most of the European freshwater flows to the BLS ($398\pm70$ km$^3$ yr$^{-1}$; it represents more than 50% of the European discharge), where the E-P balance allows for an equal contribution to the Mediterranean Sea budget though the Bosporus Strait input. Considering the Nile discharge, the closure optimization increase a lot the discharge value (from $19\pm6$ to $76\pm30$ km$^3$ yr$^{-1}$). The new value is higher to what has been monitored by GRDC near the delta (El Ekhase) in the final reported period ($59\pm30$ km$^3$ yr$^{-1}$). Recent discussions on the Nile discharge can be found in Jordà et al. (2017). Our new discharge estimate includes the groundwater discharge passing through the aquifers.

After closure optimization, the annual precipitation, evapotranspiration and moisture divergence over European drainage area are estimated to be: $2,760\pm103$, $2,151\pm102$ and $-540\pm103$ km$^3$ yr$^{-1}$ respectively. Europe accumulates most of the moisture coming from the Mediterranean Sea ($1,787\pm200$ km$^3$ yr$^{-1}$) while the Black Sea poorly evacuates its moisture towards land ($91\pm60$ km$^3$ yr$^{-1}$). Over land the contribution of the African part to the global moisture divergence is small ($87\pm14$ km$^3$ yr$^{-1}$ mainly due to the presence of the mountain Atlas). The two netflow estimates at Bosporus Strait are very close, with a difference lower than its associated uncertainty in Fig. 6. Freshwater inputs at the two Mediterranean Straits (Bosporus and Gibraltar) compensate the very large evaporation loss ($3,372\pm88$ km$^3$ yr$^{-1}$) occurring in the Mediterranean Sea. This process represents more than twice the precipitation ($1,499\pm102$ km$^3$ yr$^{-1}$).

Fig. 6 represents the whole water cycle over the region of interest with its main feature: the role of the Mediterranean Sea as the moisture and energy reservoir for the surrounding land; the poor contribution of the African coast in term of water resource, and the role of the Black Sea as the buffer process for the freshwater input. This quasi-triangular process emphasizes





the hydrological link between the surrounding land and the two Seas.

## 5.2 Comparison of the Mediterranean fluxes estimates with literature

Table 6 summarizes the comparison of the various estimates of the water fluxes in the current analysis with what can be found
in the literature. The various annual mean estimates are based on different time periods and comparison must be taken with
caution since some variability is likely to be due to the change in hydrologic regime. Sanchez-Gomez et al. (2011) focused on
the Mediterranean Sea heat and water budget using an ensemble of ERA-40-driven high resolution Regional Climate Models
(RCMs) from the FP6-EU ENSEMBLE database. The atmospheric budget was not considered in Sanchez-Gomez et al. (2011)
and no moisture divergence estimate was provided. For comparison purposes, we decided to select the RCM ensemble-mean
estimate and two particular models: the Danish HIRHAM (Hesselbjerg Christensen and Meteorologisk Institut, 1996) and the
Canadian CRCM (Plummer et al., 2006). These two models have been selected since their $E - P$ estimates are the extremes
of the RCMs ensemble. In Sanchez-Gomez et al. (2011), the netflow at Gibraltar was estimated as the deficit term of the WC:
$Gib = E - P - R - Bos$.

Mariotti et al. (2002) analyzed the WC over the Mediterranean region in the context of the NAO teleconnection over the
1979-1993 period using two reanalyses (ERA-40 and NCEP-NCAR) for precipitation, evaporation and moisture divergence.
They used the discharge data from the monitored rivers through the Mediterranean Hydrological Cycle Observing System
(MED-HYCOS) and GRDC. Their estimate includes a total Mediteranean input of 100 mm.yr$^{-1}$ from MED-HYCOS and the
Bosporus input of 75 mm.yr$^{-1}$ from the literature (Lacombe and Tchernia 1972). Mariotti et al. (2002) estimated the netflow at
Gibraltar as the balance of the Mediterranean water deficit using the equation $Gib = Div - R - Bos$ coming from the oceanic
and atmospheric budgets and the null assumptions about the storage change. Mariotti et al. (2002) used old versions of the
reanalyses and some remarks have already been raised on the precipitation and evapotransiration estimates for these versions.
Nevertheless, from our knowledge, Mariotti et al. (2002) was the last effort to estimate the WC over the Mediterranean consid-
ering the atmosphere.

Jordà et al. (2017) reviewed the state-of-the-art in the quantification of the various water component estimates. Their estimates
presented in Table 6 are the best consensual values among the scientific community. They are based on several studies and take
into account the results of Mariotti et al. (2002) and Sanchez-Gomez et al. (2011) for example. In particular, the mean Gibraltar
netflow estimate from (Jordà et al., 2016) has been commented and and new mean is provided in Jordà et al. (2017).

Table 6 also shows the results from Rodell et al. (2015) before and after their satellite data optimization based on a variational
assimilation at the annual scale. The constraint of the fluxes over the Mediterranean Sea and the Black Sea were made inde-
pendently (considering no netflow at Bosporus strait). The Mediterranean Sea was closed with no exchange to the Atlantic at
Gibraltar (no netflow). Rodell et al. (2015) provided no explicit discharge for the Mediterranean Sea but only for the Eurasian
continent.

For the four mentioned articles, only the Mediterranean Sea without the black Sea is considered. No estimate from SW+PF
methodology is provided in Table 6. Our integrated dataset is the only one to use direct observations for the netflow at Gibraltar





and to compute the Bosporus's via a WC budget. For all estimates, Table 6 presents the associated variability. While the variability of real product is computed as the standard deviation of annual values, the variability associated with the RCM mean is the inter-model spread (i.e. closer to an uncertainty estimate).

5    *Evaporation* - The RCM ensemble mean for the annual evaporation is 1,254 mm.yr$^{-1}$ with an inter-model spread of 164 mm.yr$^{-1}$. Some RCM evaluated higher annual evaporation as HIRHAM that estimated $1,377\pm55$ mm.yr$^{-1}$. On the contrary, Mariotti et al. (2002) found comparatively low evaporation with the reanalyses (1,113 and 934 mm yr$^{-1}$ with respect to NCEP and ERA). Rodell et al. (2015) estimated much higher evaporation and higher annual variability with an mean annual value of $1,391\pm157$ mm.yr$^{-1}$ using only OAFlux and $1,420\pm109$ mm.yr$^{-1}$ after optimization. Our unconstrained SW solution 10  gives an annual value of $1,300\pm34$ mm.yr$^{-1}$ and our constrained INT product gives $1,295\pm33$ mm.yr$^{-1}$. The CAL estimate is close to INT.

   *Precipitation* - The RCM ensemble mean for the annual precipitation was $442\pm84$ mm yr$^{-1}$ which is quite close to the NCEP reanalyses value in Mariotti et al. (2002). Satellite estimates in both Rodell et al. (2015) and the current study indicate 15  higher precipitation: from 576 to 571 mm.yr$^{-1}$ in (Rodell et al., 2015) after optimization and from 573 to 577 mm.yr$^{-1}$ in this work after the closure constraint. SW, INT and CAL products present similar precipitation estimates at the annual scale due to the quite low uncertainty associated with the precipitation during the optimization. Even if the spread among the RCMs was lower than for the evaporation, some RCMs as CRCM did compute even larger precipitation than what have been retrieved from satellites ($606\pm80$ mm.yr$^{-1}$). Sanchez-Gomez et al. (2011) had already noted that gauges-calibrated satellite datasets 20  such as GPCP and TMPA tend to give higher precipitation values than what was simulated in the RCMs. Precipitation over the Sea is a sensitive variable and its validation is difficult due to the lack of buoyes for *in situ* measurements. The ERA reanalyses value in Mariotti et al. (2002) was low compared with the NCEP estimate.

   *Evaporation minus Precipitation* - Sanchez-Gomez et al. (2011) focused on the $E-P$ budget to assess the physic consistency 25  in the RCM. They assumed that a model having a high evaporation tends to have a high precipitation. The average $E-P$ budget among the RCM was $812\pm180$ mm.yr$^{-1}$ and the range was between $952\pm80$ (HIRHAM model) and $602\pm107$ mm.yr$^{-1}$ (CRCM model). The inter-model spread was high for the $E-P$ budget stressing the difficulties to provide realistic water budget evaluation. Rodell et al. (2015) found similar $E-P$ budget but the associated variability was high too due to the uncertainty in evaporation. Our $E-P$ estimates are respectively $726\pm57$ and $719\pm60$ mm.yr$^{-1}$ before and after the closure 30  constraint. These values are lower but still in the RCM ensemble range. They are closer to what Mariotti et al. (2002) found with NCEP reanalyses. Jordà et al. (2017) consider the net surface flux to be $900\pm200$ mm yr$^{-1}$ which is in good agreement with the CRCM model estimate. Rodell et al. (2015) found similar $E-P$ budget but with far higher evaporation estimate which seemed quite unrealistic. Furthermore their closure constraint tends to increase the evaporation value and then the $E-P$ budget.





*Discharge* - Only the RCMs providing the runoff have been used to compute the annual value of $R$ ($124\pm46$ mm.yr$^{-1}$) in Sanchez-Gomez et al. (2011). Mariotti et al. (2002) found comparable values for the discharge, considering only the monitored rivers. Rodell et al. (2015) did not include explicit discharge into the Mediterranean Sea since the closure was done at the global scale (Eurasian continent) and no value was provided for Mediterranean freshwater input. Our discharge estimate is increased

from $144\pm21$ in SW to $155\pm15$ mm.yr$^{-1}$ in INT after the optimization. This increase is mainly driven by the re-evaluation of the Nile discharge that present larger discharge ($76$ km$^3$.yr$^{-1}$) after closure. All these discharge estimates are lower than the value prescribed in Jordà et al. (2017) ($200\pm10$ mm.yr$^{-1}$).

*Black Sea discharge* - The RCM ensemble-mean value for the freshwater input through the Bosporus strait was $87\pm60$ mm.yr$^{-1}$

stressing the high discrepancies among the RCMs. Rodell et al. (2015) closed independently the Mediterranean and the Black Sea, with no exchange between the two oceanic basins (i.e. the netflow equals to zero). In the current approach, the Black Sea discharge is computed as the deficit in the water budget for the Mediterranean Sea, in considering the netflow at Gibraltar (Gib) corrected from Jordà et al. (2016): $Bos = E - P - R - Gib$. The SW product presents unrealistic value for the Black Sea discharge ($2.0\pm615$ mm.yr$^{-1}$), this is mainly due to the high uncertainty associated to the netflow at Gibraltar. On the

other hand, the closure constraint improves the Bosporus netflow estimate which equals $129\pm60$ mm.yr$^{-1}$ with INT, after optimization. The value is close to the deficit of the Black Sea water budget (computed after optimization): $132\pm60$ mm.yr$^{-1}$ (not shown in Table 6) stressing the consistency between the two Seas water budget. The value is still higher than the estimate of $75$ mm.yr$^{-1}$ in Mariotti et al. (2002).

*Gibraltar netflow* - Rodell et al. (2015) considered no flow at Gibraltar when closing the Mediterranean WC and then pro- vided no estimate for this variable. Both Sanchez-Gomez et al. (2011) and Mariotti et al. (2002) evaluated the netflow by closing the WC over the Mediterranean region but they used different assumptions and equations. The estimate in Sanchez- Gomez et al. (2011) is based on the oceanic closure while it is based on both the oceanic and atmospheric closure in Mariotti et al. (2002). The RCM ensemble mean was $540\pm150$ mm.yr$^{-1}$ in Sanchez-Gomez et al. (2011), while Mariotti et al. (2002)

found lower value with the reanalyses ($493$ and $370$ mm.yr$^{-1}$ with NCEP and ERA). Jordà et al. (2017) give two values for the netflow at Gibraltar: one from direct observations but suffering from large uncertainties ($850\pm400$ mm.yr$^{-1}$), and the other as the deficit of the water budget ($600\pm200$ mm.yr$^{-1}$). The value in INT and CAL estimate are impacted by the closure constraint. The netflow estimate after optimization ($428\pm124$ mm.yr$^{-1}$) is lower than what can be found in Jordà et al. (2017) but in the range of the RCM water budget deficit.

*Moisture divergence* - No moisture divergence was provided by the RCMs in Sanchez-Gomez et al. (2011). Mariotti et al. (2002) found moisture divergence to be $659$ mm yr$^{-1}$ in NCEP and $488$ mm.yr$^{-1}$ in ERA. Rodell et al. (2015) estimated the divergence to be $848\pm105$ mm.yr$^{-1}$ after optimization. The difference between Rodell et al. (2015) or Mariotti et al. (2002) estimates and what is found in the current study is mainly driven by the discrepancy between the three reanalyses: Modern-Era

Retrospective Analysis for ReSearch and Applications (MERRA) used by Rodell et al. (2015), NCEP and ERA-40 in Mariotti



et al. (2002), and ERA-I used in the current analysis. Recent works focusing on atmospheric reanalyses comparisons have demonstrated the ERA-I quality. Stopa and Cheung (2014) have stressed the ERA-I performances in the representation of long term wind variability, critical for the representation of moisture divergence. Brown and Kummerow (2013) have pointed out that satellite derived $E - P$ (SeaFlux- GPCP) correlates well with ERA-Interim atmospheric moisture divergence. Trenberth

et al. (2011) have assess the performance of ERA-I reanalysis for atmospheric moisture budgets consideration.

## 6   Conclusions

The main goal of this work was to build a multi-component dataset describing the water cycle by constraining the WC closure. Various methodologies have been presented and particular attention has been put on the INTegration method. This approach full-fills the previous stated objectives: being a pixel-wise dataset but in which the WC closure is controled. INT is an integrated

dataset that shows several benefits compared with previous studies. The INT product allows to reduce the RMS of the WC budget residual down to 3.55 mm/month over land and 5.27 mm/month in the atmosphere. These reductions represent an improvement of respectively 78% and 80% compared with the best un-constrained satellite combination dataset. The temporal coverage of INT is limited by the common coverage period 2004-2009 of all the satellite estimates used in this study (see Table A1).

The INT dataset has been evaluated at various scales. Even if the evaluation is a difficult task and the presented work is not exhaustive, our results show that the consideration of the WC closure allows to reduce differences with the *in situ* measurements. At the sub-basin scale, the overall precipitation is closer to the *in situ* gridded EOBS dataset after being constrained. The Sea Water Level estimate is also improved compared to the altimetry estimate. At the pixel scale, the INT estimate shows a better agreement with *in situ* tower measurements from the FLUXnet2015 database.

The WC has been analyzed in terms of long-term means over the 2004-2009 period and compared with previous literature. The INT methodology has improved estimates of the Mediterranean water components. The INT product provides more realistic values for both Bosporus and Gibraltar netflows by constraining it with the satellite observations. Note that the estimate of the Bosporus is mainly driven by the Gibraltar estimate and can then be improved as the Gibraltar netflow evaluation would become more accurate.

This study conducted on the Mediterranean Sea is innovative from previous work. The Mediteranean WC has already been well investigated by Mariotti et al. (2002) and Sanchez-Gomez et al. (2011) relying on models and reanalyses. At global scale, Rodell et al. (2015) close independently the Mediterranean and Black Seas using satellite observation while Sanchez-Gomez et al. (2011) close the Mediterranean Sea WC in estimating the Gibraltar netflow as the WC budget deficit. This study aims at providing a full description of the WC, based on fewer hypothesis. It is the first effort to close the WC, at the surface and in the

atmosphere over the whole Mediterranean region, using satellite observations and *in situ* measurement for Gibraltar netflow.

   There are still uncertainties concerning the observation of the water cycle and its components but the INT methodology is the best effort for including coherency among these independent component estimates. This multiple-component dataset shows promising aspect for forcing, calibrating or constraining regional models with water conservation (as required by the





community). The two databases can be freely obtained under request to the corresponding author.

Besides the capability to introduce WC closure coherency with our INT approach, the current work has introduced also the CAL product which is a calibration of the satellite products that can be used to extrapolate in time the closure constraint. The

5 CAL product is less efficient to close the WC but presents the advantage to have longer temporal coverage.

*Acknowledgements.* We would like to thank the ESA (European Space Agency) and its Support To Science Element program for funding the "Water Cycle Observation Multi-mission Strategy For Mediterranean region" project (ESRIN Contract No. 4000114770/15/I-SBO; wac-mosmed.estellus.fr). We are grateful for the E-OBS dataset from the EU-FP6 project ENSEMBLES (http://ensembles-eu.metoffice.com), and the data providers of the ECA&D project (http://www.ecad.eu). We would like to thank the WACMOS-Med partners for the interesting

10  related discussions and Phillipe Drobinsky and Véronique Ducrocq for their support to WACMOS-Med and his role in the HYMEX project.





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

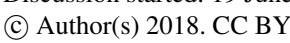


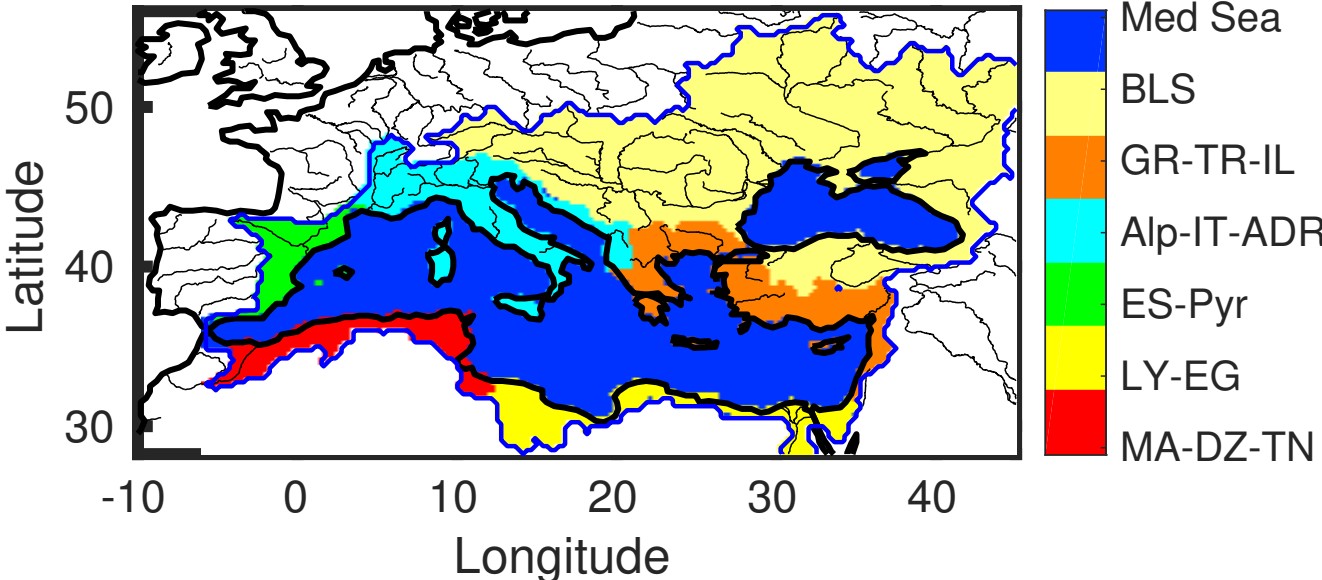

**Figure 1.** Region of interest. Sub-basins have been computed using a hydrological model (Wu et al., 2011), and rivers are from HydroShed (http://www.hydrosheds.org/). See text for the definition of the sub-basins.





**Figure 2.** Steps of the spatialisation of the budget closure for the INT solution, from the SW to the INT solutions: Precipitation (left) and evapotranspiration (right), for July 2008. Units are in mm/month.




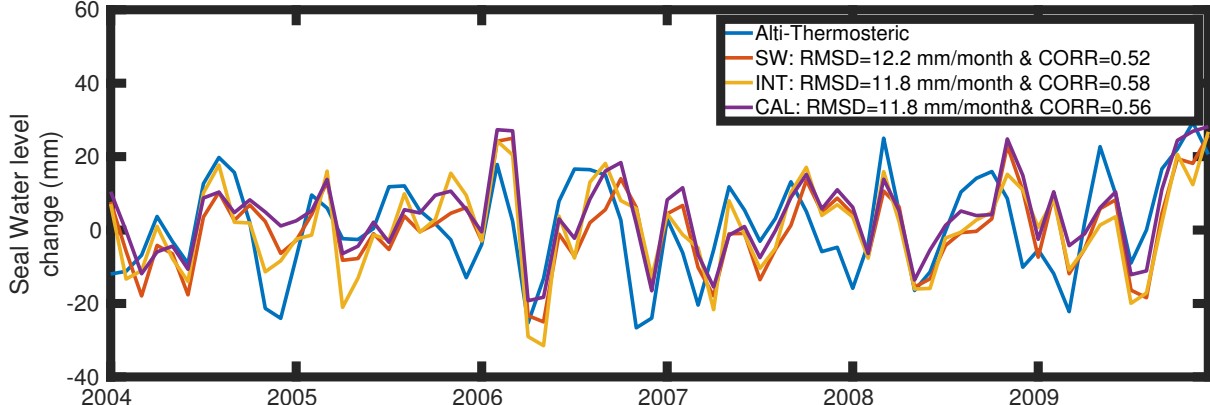

**Figure 3.** Sea water level evaluation of SW, INT and CAL estimates compared to altimetry minus thermal content.





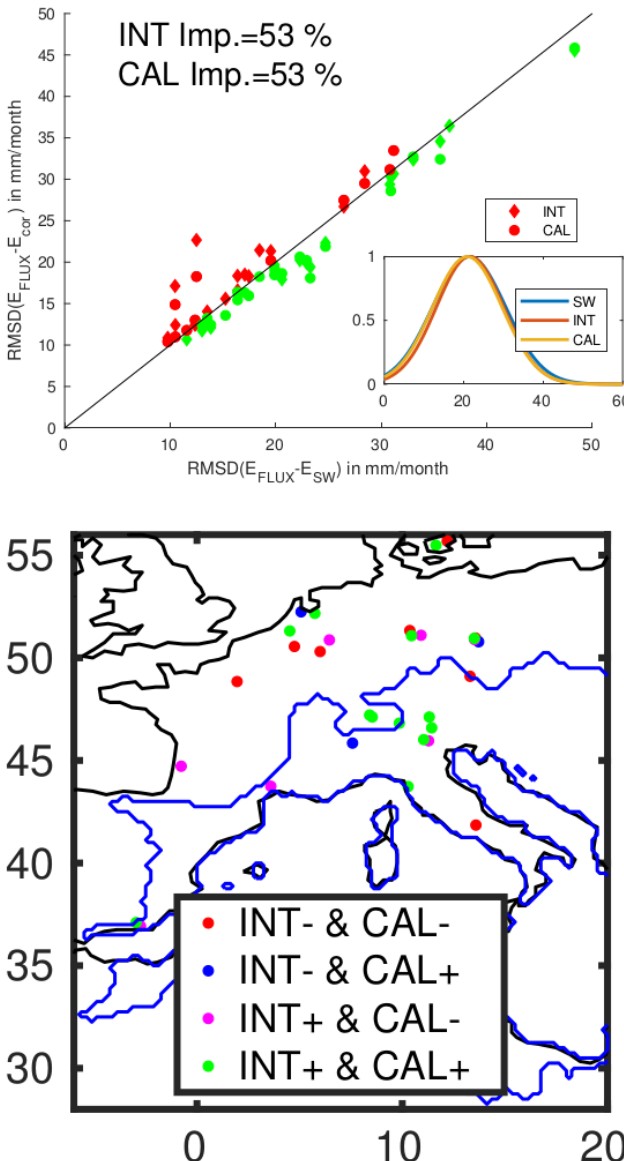

**Figure 4.** Top panel: Scatterplot of the RMSD between FLUXnet station and the SW, INT and CAL products, for evapotranspiration. Dots under the 1:1 line (green) show improvement, and dots over the line (red) show degradation. The encapsulated figure shows the distribution of the differences with the Fluxnet estimates. Bottom panel: Location of the FLUXnet stations used for validation: green dots show an improvement for INT and CAL compared to SW, yellow dots show improvement only for CAL, and blue only for INT. Red dots is where no improvement is observed. Blue line bounders the total basin area.





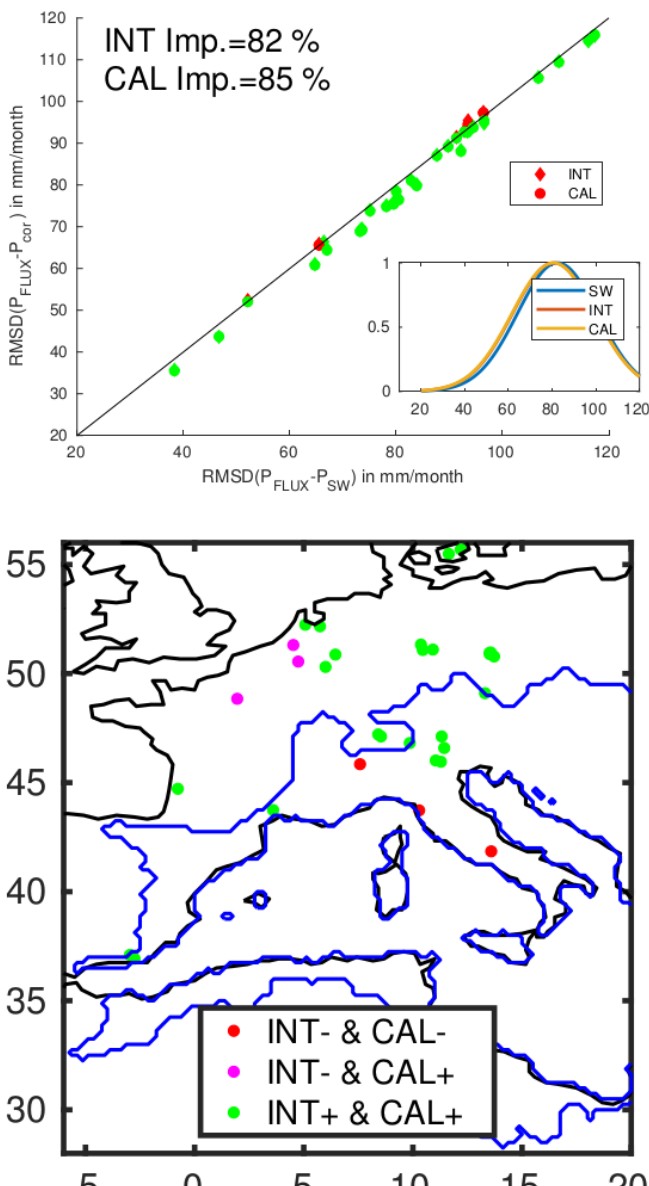

**Figure 5.** same as Fig. 4 but for precipitation.





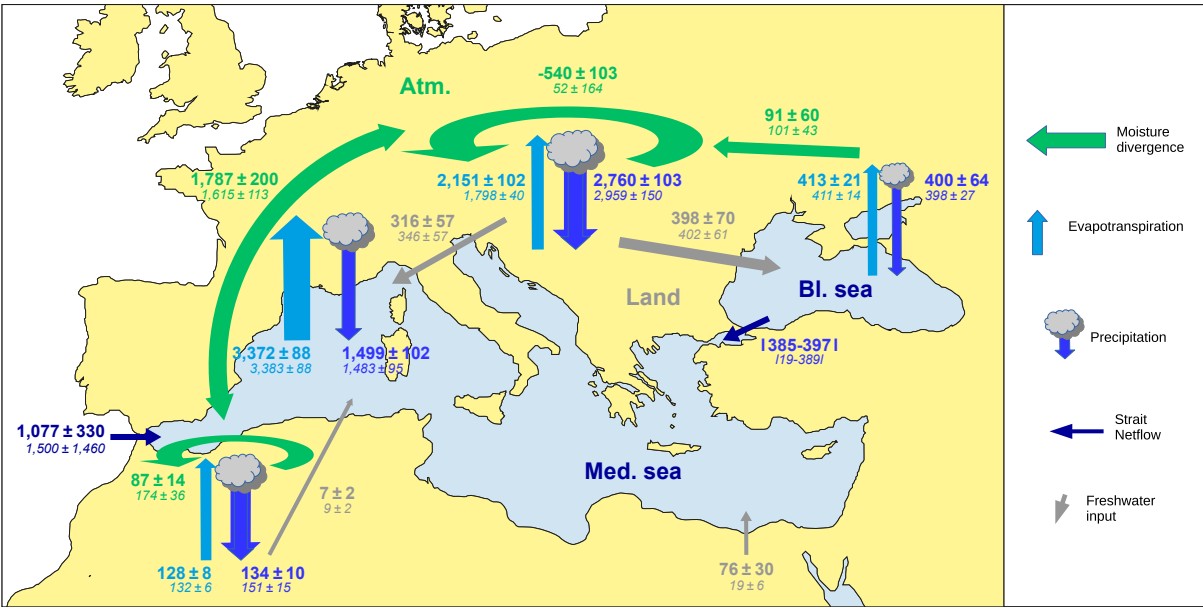

**Figure 6.** Mean annual fluxes (km$^3$ yr$^{-1}$) of the Mediterranean water cycle and associated uncertainties in SW (small font) and INT (large font) during the 2004-2009 period.




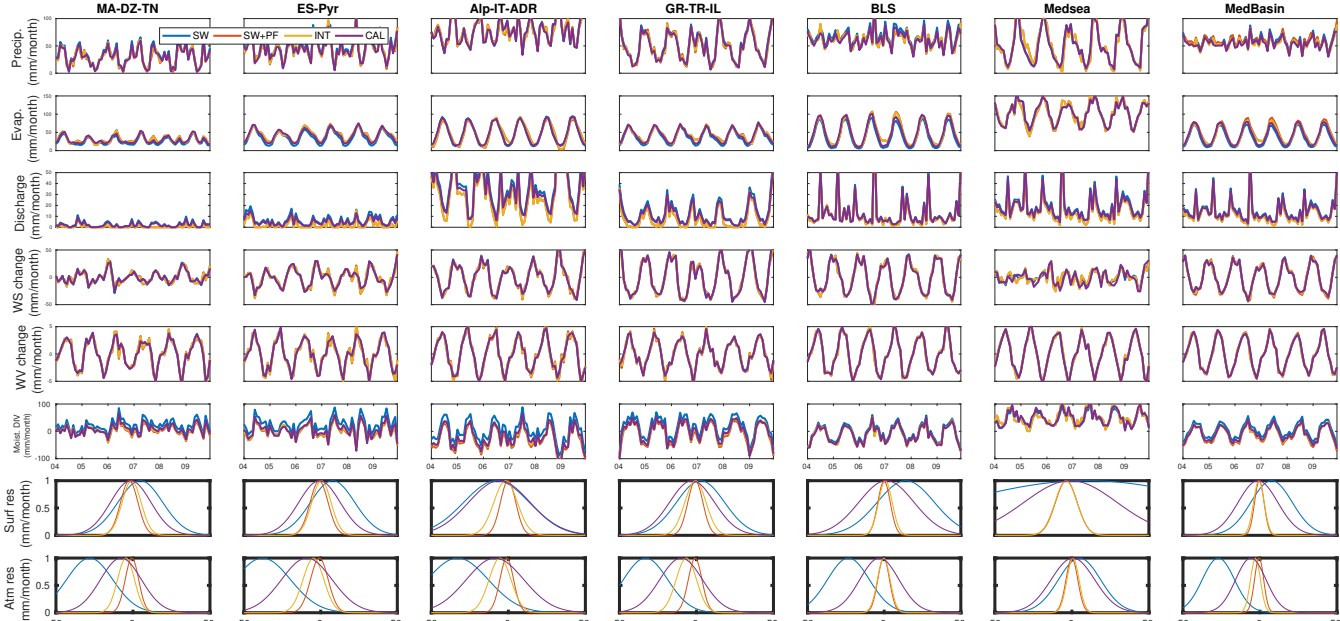

**Figure A1.** Comparison of the six water components estimates and the pdf of the two WC budget error (in row). The estimates are for the 6 terrestrial sub-basins, the oceanic part and the total land (in column) through the various methodologies presented in the study: SW, SW+PF, INT and CAL.





| | **Princeton** | **NASA NEWS** | **ESA** |
|---|---|---|---|
| Integration method | Simple Weighting + CKF for budget closure | Variational Assimilation | Simple Weighting + PF for budget closure |
| References | Pan and Wood (2006) Sahoo et al. (2011); Pan et al. (2012) | Rodell et al. (2015) L'Ecuyer et al. (2015) | Aires (2014); Munier et al. (2014) Munier and Aires (2017) |
| Strategy | Assimilation with VIC model | Fluxes optimization | Fluxes optimization |
| Source | model +observations | model+ observations | observations |
| Budget | Terrestrial WC only | Terrestrial, oceanic & atmospheric WC | Terrestrial, **oceanic & atmospheric WC** |
| Spatial scale | basin[1] | continent | pixel to basin scale |
| Multiplicity of datasets | yes weighted average | only for $E$ | yes weighted average |
| Uncertainty reference | gauges density & average product | average product | prescribed (literature) |
| Spatial multi-scaling | no | yes: dependent continents through one ocean | **yes: simultaneously at basin and sub-basins scales** |
| Temporal multi-scaling | no: monthly | no: annually + interpolation[2] | **yes: monthly & annually** |
| State vector | $X_T=[P_l \ E_l \ R_l \ \Delta S_l]^t$ | $F=[P \ E \ R \ Div]^t$ $Res=[\Delta S \ \Delta W]^t$ | $X_l=[P_l \ E_l \ R_l \ \Delta S_l \ \Delta W_l \ Div_l]^t$ **over land** $X_o=[P_o \ E_o \ \Delta S_o \ Gib]^t$ **over Sea** $X_{lo}=[X_l \ X_o]$ **for both** |
| Uncertainties | $B_T$ is the error covariance of $X_T$ | $S_{Res}$ and $S_F$ error covariance matrices | $B_{lo}$ is the error covariance of $X_{lo}$ |
| Model | $G_T=[1,-1,-1,-1]$ | $A$: Matrix of budgets[3] | $G_l=\begin{bmatrix} 1 & -1 & -1 & -1 & 0 & 0 \\ -1 & 1 & 0 & 0 & -1 & -1 \end{bmatrix}$ $G_o=\begin{bmatrix} 1 & -1 & -1 & 0 & 0 & -1 \\ -1 & 1 & 0 & -1 & -1 & 0 \end{bmatrix}$ $L_{lo}=\begin{bmatrix} 0 & 0 & \frac{A_{land}}{A_{Sea}} & 0 & 0 & 0 \\ 0 & 0 & 0 & 0 & 0 & 0 \end{bmatrix}$ $G_{lo}=\begin{pmatrix} G_l & 0 \\ L_{lo} & G_o \end{pmatrix}$ |
| Closure equation | $G_T \cdot X_T = 0$ | $Res = A \cdot F$ | $G_{lo} \cdot X_{lo} = \mathbf{r}, \ \mathbf{r} \sim \mathcal{N}(\mathbf{0}, \sum)$ **with $\sqrt{\sum}$=2 mm/month** |
| Type of constraint | strong constraint | strong constraint + Interpolation | **relaxed constraint** |
| Closure solution | $X_{Tc}=X_T+K_T \cdot (0 - G_T X_T)$ with $K_T = B_T G_T \cdot (G_T B_T G_T^t)^{-1}$ | $F_c = F + Q^{-1} J^t S_{Res}^{-1}(Res - AF)$ $J$ the Jacobian of $Res$ w/r to $F$ and $Q = (J^t S_{Res}^{-1} J + S_F^{-1})^{-1}$ | $X_{loc} = (I - K_{PF} G_{lo} \sum^{-1} G_{lo}^t) \cdot X_{lo}$ $K_{PF} = (B_{lo}^{-1} + G_{lo} \sum^{-1} G_{lo}^t)^{-1}$ |

**Table 1.** The three main initiatives for budget closure constraint and their technical differences. [In the third column, bold font indicates the new features of the methodology presented in this article]. Subscript are: $l$ for land, $o$ for ocean, both include the atmosphere. All notations are summerized in Table A1. (1) Zhang et al. (2016) recently developed a WC-VIC assimilation scheme at the $0.5°$ pixel scale; (2) Rodell et al. (2015) used a two-step integration methods with annual closure simply downscaled at the monthly scale, plus a Lagrange interpolation for closure relaxation; (3) see Rodell et al. (2015) for details and hypothesizes.



| | Zhang et al. (2017) | Sahoo et al. (2011) | Munier et al. (2014) | Rodell et al. (2015) | Our study |
|---|---|---|---|---|---|
| Area | Europe | Danube basin | Mississippi basin | Eurasia | Med. region |
| P | - | - | 5 mm/month | 3 mm/month | 4 mm/month |
| | 36% | 47% | 37% | 24% | 25% |
| E | - | - | 5.8 mm/month | 5 mm/month | 6 mm/month |
| | 41% | 32% | 49% | 65% | 55% |
| R | - | - | 1 mm/month | 3 mm/month | 2 mm/month |
| | 7% | 3% | 1.5% | 11% | 6% |
| $\Delta$S | - | - | 2.9 mm/month | - | 3 mm/month |
| | 14% | 18% | 12.5% | - | 14% |

**Table 2.** Comparison of the uncertainty specifications for terrestrial water components. The weights associated to a variable (computed as the ratio between the particular variable uncertainty with respect to the sum of all the uncertainties $\sigma_i^2 / \sum_i \sigma^2$) are expressed in percentage.





|        | EO merging | Spatial resol. | Coverage period* | WC budget closure |
|--------|------------|----------------|------------------|-------------------|
| OS     | no         | pixel          | 1993-2012        | - -               |
| SW     | yes        | pixel          | 1980-2012        | -                 |
| SW+PF  | yes        | basin scale    | 2004-2009        | ++                |
| INT    | yes        | pixel          | 2004-2009        | ++                |
| CAL    | yes        | pixel          | 1980-2012        | +                 |

**Table 3.** Main characteristics of the five merging methods in this study: EO stands for Earth Observation satellite datasets, and * means not considering the GRACE period coverage. The last column shows the capability of the methodology to close the WC budget. '- -' means bad closure, '-' means quite bad closure, '+' means quite good closure and '+ +' means good closure.





| | | Climatic sub-basins | | | | | | | | | | LAND | | OCEAN | |
|---|---|---|---|---|---|---|---|---|---|---|---|---|---|---|---|
| | | MA-DZ-TN | | ES-Pyr | | Alp-IT-ADR | | GR-TR-IL | | BLS | | | | | |
| | | surf | atm | surf | atm | surf | atm | surf | atm | surf | atm | surf | atm | surf | atm |
| ERA-I | | 34.3 | 15.3 | 37.8 | 18.1 | 31.2 | 13.7 | 30.6 | 12.0 | 18.0 | 8.0 | 13.6 | 13.8 | 86.7 | 6.2 |
| OS | | 25.1 | 36.0 | 27.5 | 43.5 | 28.5 | 37.7 | 25.8 | 39.7 | 25.4 | 27.3 | 19.8 | 15.1 | 75.2 | 24.7 |
| SW | | 18.2 | 31.8 | 17.5 | 40.7 | 21.5 | 38.3 | 17.6 | 35.6 | 25.1 | 26.5 | 16.6 | 16.6 | 74.3 | 15.7 |
| SW+PF | | 4.46 | 3.04 | 4.38 | 3.99 | 4.42 | 3.07 | 4.46 | 3.21 | 3.64 | 2.82 | 2.78 | 2.28 | 7.18 | 3.13 |
| | | 75% | 90% | 74% | 90% | 79% | 91% | 74% | 90% | 85% | 89% | 83% | 91% | 91% | 80% |
| INT | | 5.23 | 5.82 | 5.15 | 6.47 | 7.70 | 7.65 | 6.62 | 8.16 | 4.21 | 3.20 | 3.79 | 4.07 | 7.18 | 3.13 |
| | | 71% | 81% | 70% | 84% | 64% | 80% | 62% | 77% | 83% | 87% | 77% | 84% | 91% | 80% |
| CAL | | 13.14 | 14.48 | 13.38 | 17.77 | 20.13 | 20.21 | 14.51 | 16.77 | 18.00 | 13.03 | 12.79 | 11.44 | 24.63 | 12.50 |
| | | 27% | 54% | 23% | 56% | 6% | 47% | 17% | 52% | 28% | 50% | 22% | 56% | 66% | 17% |

**Table 4.** RMS of the WC budget residual (in mm/month) over the sub-basin using OS,SW,SW+PF,INT and CAL solution and for the period 2004-2009. Percentage of improvement of the RMS of the residuals from SW solution to constrained methods is also shown. For comparison purpose, result using ERA-I dataset is also depicted.





| | | Climatic sub-basins | | | | | Continental |
|---|---|---|---|---|---|---|---|
| | | MA-DZ-TN | ES-Pyr | Alp-IT-ADR | GR-TR-IL | BLS | |
| Correlation | SW | 0.81 | 0.88 | 0.87 | 0.87 | 0.79 | 0.78 |
| | SW+PF | 0.84 | 0.90 | 0.88 | 0.87 | 0.81 | - |
| | INT | 0.84 | 0.89 | 0.88 | 0.87 | 0.81 | 0.80 |
| | CAL | 0.81 | 0.88 | 0.87 | 0.87 | 0.79 | 0.79 |
| RMSD | SW | 14.01 | 16.69 | 21.78 | 23.04 | 20.56 | 15.68 |
| | SW+PF | 13.60 | 14.10 | 22.42 | 21.98 | 16.64 | - |
| | | 2% | 15% | -3% | 4% | 19% | - |
| | INT | 13.59 | 14.35 | 21.88 | 21.83 | 16.84 | 12.93 |
| | | 2% | 14% | -1% | 5% | 18 % | 17% |
| | CAL | 14.00 | 14.83 | 22.06 | 21.64 | 17.23 | 13.16 |
| | | 0% | 11% | -2% | 6% | 16% | 16% |

**Table 5.** Comparison of the SW, SW+PF, INT and CAL precipitation solutions with the EOBS dataset, in terms of correlations, RMSD, and percentage of improvement of the RMSD compared to the SW solution.





| References | | E | P | E-P | R | Bos | Gib | Div |
|---|---|---|---|---|---|---|---|---|
| Sanchez-Gomez et al. (2011) | HIRHAM | 1,377±55 | 425±57 | 952±80 | 116±30 | 116±30 | 720±100 | - |
| 1957-2002 | MEAN | 1,254(±164) | 442(±84) | 812(±180) | 124(±46) | 87(±60) | 540(±150) | - |
| | CRCM | 1,208±72 | 606±80 | 602±107 | 73±40 | 110±50 | 420±130 | - |
| Mariotti et al. (2002) | NCEP | 1,113 | 433 | 680 | 100 | 75 | 494 | 659 |
| 1979-1993 | ERA-40 | 934 | 331 | 603 | 100 | 75 | 370 | 488 |
| Jordà et al. (2017) | Prescribed | - | - | 900±200 | 200±10 | 100±20 | 850±400 | - |
| 2005-2010 | values | | | | | | *600±200* | |
| Rodell et al. (2015) | orginal | 1,391±157 | 576±76 | 815±157 | - | 0 | 0 | 866±131 |
| 2000-2010 | optimized | 1,420±109 | 571±73 | 849±109 | - | 0 | 0 | 848±105 |
| Current study | SW | 1,300±34 | 573±36 | 726±57 | 144±21 | 2±615 | 575±561 | 620±44 |
| 2004-2009 | INT | 1,295±33 | 577±40 | 719±60 | 155±15 | 129±60 | 428±124 | 677±77 |
| | CAL | 1,295±34 | 574±36 | 721±57 | 155±20 | 80±250 | 428±196 | 680±53 |

**Table 6.** Comparison in the literature for the Mediterranean Sea (without the Black Sea) average annual mean fluxes and their associated variability (in mm yr$^{-1}$). While the variability of real product is computed as the standard deviation of annual values, the uncertainty associated with the Regional Climate Models mean is the inter-model spread. The period of analysis for the various studies are recalled.



| Mathematical symbols | |
|---|---|
| $M^t$ | Transpose |
| $\Delta M$ | Differenciation |
| $\frac{\delta M}{\delta t}$ | Derivative |
| $N$ | Normal distribution |
| $\sigma$ | Standard deviation |
| RMS | Root Mean Square |
| RMSD | Root Mean Square of the Difference |
| **Subscript** | |
| $M_T$ | Terrestrial |
| $M_l$ | Over land (terrestrial plus atmospheric) |
| $M_l^i$ | Over the $i^{th}$ sub-basin (terrestrial plus atmospheric) |
| $M_o$ | Over ocean(oceanic plus atmospheric) |
| $M_{lo}$ | Global: land + ocean |
| $M_c$ | Constrained |
| $M_{SW}$ | Estimate through SW merging technique |
| $M_{PF}$ | Estimate through SW+PF approach |
| $M_{INT}$ | Estimate through INT approach |
| **Water components** | |
| $P$ | Precipitation |
| $E$ | Evapotranspiration |
| $S$ | Water storage |
| $W$ | Precipitable water |
| $Div$ | Vertically integrated Moisture divergence |
| $Gib$ | Gibraltar oceanic netflow |
| $Bos$ | Bosporus oceanic netflow |
| **WC State vector and associated uncertainty matrices** | |
| $X_T,\ B_T$ | Terrestrial state vector |
| $X_l$ | Water cycle state vector over land (within the atmospheric aspect) |
| $X_l^{(i)}$ | Water cycle state vector over the $i^{th}$ sub-basin (terrestrial plus atmospheric) |
| $X_o$ | Water cycle state vector over Sea (within the atmospheric aspect) |
| $X_{lo}, B_{lo}$ | Gobal water cycle state vector |
| $X_{lo}^{Month}$ | Gobal water cycle state vector for a particular month |
| $r,\ \sum$ | Tolerated WC budget residuals |
| **Closure matrices** | |
| $G_T$ | Terrestrial budget |
| $G_l$ | WC closure over land (within the atmospheric closure) |
| $G_l^{(i)}$ | Water cycle closure over the $i^{th}$ sub-basin (terrestrial plus atmospheric) |
| $G_o$ | Water cycle closure over Sea (within the atmospheric aspect) |
| $G_{lo},$ | Gobal water cycle closure |
| $L_{lo},$ | Freshwater equality between land and Sea |
| $A_{land}$ | Total drainage area of the Mediterranean Sea within the Black Sea |
| $A_l^{(i)}$ | Drainage area of the $i^{th}$ sub-basin |
| $A_{Sea}$ | Sea area |
| $L_{lo}^{(i)},$ | Freshwater equality between the $i^{th}$ sub-basin and Sea |
| $GA_{lo}$ | Global water cycle closure for all the month within the year |
| $N_{lo}$ | Modified version of $G_{lo}$ |
| $N_l^{(i)}$ | Modified version of $G_l^{(i)}$ |
| **constraint filter** | |
| $K_T$ | Terrestrial constraint |
| $K_{merge}$ | Merging matrix in SW methodology |
| $K_{PF}$ | Global water cycle constraint via PF methodology |
| $K_{an}$ | Theoretical analysis filter |

**Table A1.** Notation used in this study





| Dataset | Time coverage | Spatial res. (°) | Temporal res. | Description | Producer | Reference |
|---------|---------------|------------------|---------------|-------------|----------|-----------|
| **Precipitation** | | | | | | |
| GPCP | 1979-2015 | 2.5 | daily | from multiple satellites and gauges | U. of Maryland | Adler et al. (2003) |
| CMORPH | 1998-2015 | 0.25 | 30 min | from microwave and infrared | NOAA | Joyce et al. (2004) |
| TMPA | 1998-2015 | 0.25 | 3h | from multiple satellites and gauges | NASA | Huffman et al. (2007) |
| PERSIANN | 2000-2013 | 0.25 | 3h | from microwave and infrared | CHRS | Ashouri et al. (2015) |
| ERA-I Precipitation | 1980-2015 | 0.25 | 12h | reanalysis | ECMWF | Dee et al. (2011) |
| EOBS Precipitation | 1950-2006 | 0.25 | daily | *in situ* gridded | project ENSEMBLES | Haylock et al. (2008) |
| FLUXnet precipitation | 2002-2010 | - | monthly | *in situ* | FLUXnet | Falge et al. (2017) |
| **Evapotranspiration** | | | | | | |
| GLEAM | 1980-2012 | 0.25 | daily | satellite observation, gauges and reanalysis | U. of Amsterdam and U. of Ghent | Martens et al. (2016) |
| MOD16 | 2000-2012 | 0.25 | 8 days | satellite observation | NTSG | Mu et al. (2011) |
| NTSG | 1983-2012 | 0.25 | monthly | satellite observation and reanalysis | NSTG | Zhang et al. (2010) |
| ERA-I evapotranspiration | 1980-2015 | 0.25 | 12h | reanalysis | ECMWF | Dee et al. (2011) |
| FLUXnet evapotranspiration | 2002-2010 | - | monthly | *in situ* | FLUXnet | Falge et al. (2017) |
| **Evaporation** | | | | | | |
| OAflux | 1985-2015 | 1 | daily | from satellite and reanalysis | WHOI | Sun et al. (2003) |
| Seaflux | 1998-2015 | 0.25 | 3h | from satellite, reanalysis and *in situ* | GEWEX | Curry et al. (2004) |
| ERA-I Evaporation | 1980-2015 | 0.25 | 6h | reanalysis | ECMWF | Dee et al. (2011) |
| **Water storage** | | | | | | |
| CSR | 2002-2012 | 0.25 | monthly | GRACE | CSR | Bettadpur (2012) |
| GFZ | 2002-2012 | 0.25 | monthly | GRACE | GFZ | Dahle et al. (2013) |
| GRGS (land only) | 2002-2012 | 0.25 | monthly | GRACE | CNES | Biancale et al. (2005) |
| JPL | 2002-2012 | 0.25 | monthly | GRACE | JPL | Watkins and Yuan (2014) |
| MSC-JPL | 2002-2015 | 0.25 | monthly | GRACE | JPL | (Watkins et al., 2015) |
| **Precipitable water** | | | | | | |
| Globalvapor | 1996-2015 | 0.5 | daily | merged estimates from satellite | DWD, GEWEX | Schneider et al. (2013) |
| ERA-I Wator vapor | 1979-2015 | 0.25 | 6h | reanalysis | ECMWF | Dee et al. (2011) |
| **Discharge** | | | | | | |
| CEFREM | 1980-2009 | < 0.25 | annual | *in situ* | Cefrem | Ludwig et al. (2009) |
| ORCHIDEE | 1980-2009 | 0.5 | monthly | model | LMD | Polcher et al. (1998) |
| **Moisture flux divergence** | | | | | | |
| ERA-I Moisture divergence | 1979-2015 | 0.25 | 6h | reanalysis | ECMWF | Dee et al. (2011) |
| **Gibraltar netflow** | | | | | | |
| IMEDEA- netflow | 2004-2010 | - | monthly | *in situ* & model | IMEDEA | Jordà et al. (2016) |

**Table B1.** Overview of the various datasets used in this study. Their common coverage period, on which the WC budget is estimated, is 2004-2009.