# Peer review of "Integrating multiple satellite observations into a coherent dataset to monitor the full water cycle - Application to the Mediterranean region"

_Hydrology and Earth System Sciences, 2018_

## Referee Comment (RC1) · B. Su (Referee) · 24 Jul 2018

Comments on 'Integrating multiple satellite observations into a coherent dataset to monitor the full water cycle - Application to the Mediterranean region' by Pellet et al.

The MS presents a substantial effort in integrating multiple satellite observations into a coherent data set for monitoring the water cycle of the Mediterranean basin. From a technical point of view, many data products and acronyms are introduced but the reader is rather overwhelmed by the details and misses the central message the MS is trying to convey.

[Figure]

I would suggest the following revision:

1. Since the described project and the MS aims to produce a satellite observations of water cycle, I would suggest that the logic and methods proposed for generating climate data records be followed and organized as such (see e.g. Su et al., 2018, BAMS).

2. There is a need to make sure that the used datasets are independent of each other. For example the GLEAM v3c evaporation dataset is used, but the GLEAM dataset uses also precipitation dataset as input. Could the authors check and verify the independence of such datasets?

3. In the first ESA WACMOS project (Su et al., 2014, JAG), an independent evaporation product was generated and is updated continuously. The monthly evapotranspiration for global land area from satellite data (global land 5 km spatial resolution monthly ET dataset, 2000-2017) is produced with a revised SEBS algorithm (Su et al., 2002, HESS; Chen et al. 2013, JAMC) with input as MODIS LST, NDVI, Global forest height, GlobAlbedo and meteorology from ERA-I. A recent comparison of the SEBS ET has reported by Bhattarai et al, 2018, HESS, with the MOD16 ET and a method by integrating radiometric surface temperature (TR) into the Penman-Monteith (PM) equation (STIC). The authors are advised to take a look. The data may be accessed at: http://en.tpedatabase.cn/portal/MetaDataInfo.jsp?MetaDataId=249454.

4. The authors presented statistics as a quality criteria of the WC closure. I suggest to spend some effort in checking the dynamics and physics of the different datasets. I am not sure the correlation coefficients and RMSDs are the most suitable relevant statistical criteria for spatial-temporal datasets.

5. The English needs improvement. There are lots of typos and awkward expressions.

References:

Bhattarai, N., Mallick, K., Brunsell, N. A., Sun, G., & Jain, M. (2017). Regional evap-

otranspiration from image-based implementation of the Surface Temperature Initiated Closure (STIC1. 2) model and its validation across an aridity gradient in the conterminous United States.

Chen, X., Su, Z., Ma, Y., Yang, K., Wen, J., & Zhang, Y. (2013). An improvement of roughness height parameterization of the Surface Energy Balance System (SEBS) over the Tibetan Plateau. Journal of applied meteorology and climatology, 52(3), 607-622.

Su, Z. (2002). The Surface Energy Balance System (SEBS) for estimation of turbulent heat fluxes. Hydrology and earth system sciences, 6(1), 85-100.

Su, Z., Fernández-Prieto, D., Timmermans, J., Chen, X., Hungershoefer, K., Roebeling, R., ... & Wolters, E. (2014). First results of the earth observation Water Cycle Multi-mission Observation Strategy (WACMOS). International journal of applied earth observation and geoinformation, 26, 270-285.

Su, Z., Timmermans, W., Zeng, Y., Schulz, J., John, V. O., Roebeling, R. A., ... & Swinnen, E. (2018). An overview of European efforts in generating climate data records. Bulletin of the American Meteorological Society, 99(2), 349-359.

---

## Referee Comment (RC2) · Anonymous Referee #2 · 22 Aug 2018

Review of "Integrating Multiple Satellite Observations into a Coherent Dataset to Monitor the Full Water Cycle – Application to the Mediterranean Region"

Overall, I found this to be a strong and well-written paper. It makes two (worthwhile) contributions. First, a methodological contribution regarding the appropriate approach for simultaneously enforcing water closure within land, atmosphere and ocean domains. Second, it advances the state-of-the-art in terms of water balance estimates for the Mediterranean Basin. These contributions are significant and worthy of publication in HESS. Nevertheless, there are three major points that the authors should address before final publication. I suspect that some of my concerns arise from misunderstandings on my part (rather than actual flaws) and can be addressed via re-writing for improved clarity.

MAJOR

1) The paper needs to do a better job of describing the INTegration (INT) methodology and its impact on subsequent stock and flux predictions. My understanding is that the INT approach is applied with two aims: 1) to downscale sub-basin scale results down to the pixel scale and 2) to extrapolate (balance-constrained) results OUTSIDE of the Mediterranean. This raises two questions:

a) If INT down-scales and extrapolates outside of the Mediterranean Basin – why does it have any impact on sea-level results in Figure 3 (which presumably reflect spatially averaged conditions within the Basin...neither of which are impacted by INT)? To me, it seems like SW+PF and INT should yield the same results for a sea-level metric. However, these results are used specifically to motivate the added value of INT (line 15 of page 17). Is the improvement in INT versus SW attributable to INT? Or would it also occur for SW+PF?

b) What exactly is the rational for the extrapolation portion of INT? Why would you ever want to extrapolate? Why not just apply terrestrial closure (at a minimum) to Northern Europe SW results? Also, how does this extrapolation contribute to the (bottom-line) analysis in Figure 6? I presume it facilitates the application of a larger atmospheric water balance analysis, but – given that this is a Mediterranean Basin analysis - it seems strange to extrapolate BEYOND the Mediterranean Basin. It would improve the manuscript if this extrapolation step was better motivated.

2) I feel like the paper could do a better job describing its approach to error estimation (and the effect of these estimates on its merging results).

a) I was confused by the treatment of EO uncertainty throughout the manuscript. First, in Line 8 of page 8, the manuscript says states "...we considered the same uncertainty

for all data sets of given parameter following de-biasing..." However, later in Section 3.3, it seems as if a different uncertainty is assigned to various precipitation estimates when applying Equation (5). Can these descriptions be made more consistent?

b) in Section 3.4, the authors invoke a filter-based closure constraint that varies as a function of a $\Sigma$ matrix but do not describe how this matrix was derived. If would be helpful if this was clarified.

c) Equation (5) appears to use the temporal standard deviation of individual products (precipitation products in the example given) after seasonal bias correction as a proxy for the magnitude of their random error. This seems like a dangerous assumption. Assume, for example, that you had a precipitation product that simply mimicked the TMPA seasonal climatology (used here as the de-biasing reference). Given large inter-annual variability in rainfall, this product would be a poor rainfall product to use in a water balance context. However, it would have a low temporal standard deviation, and (therefore) be heavily weighted by Equation (5).

Also - on a related point - after you de-bias the precipitation products, does it really matter (for a long-term water balance study) how - or even IF - you merge the products? After de-biasing, they all have the same long-terms means and will thus produce the same long-term water balance analysis.

3) Page 16, Line 3. The PF approach is designed explicitly to reduce closure residuals. So, it is questionable to use the reduction of closure residuals as evidence that that PF approach is "working" or that a closure constraint is necessary. Another possibility is that not all flux and stock terms are being accounted for him. That is, the flux/stock estimates utilized here are actually accurate but nevertheless should not close. Some discussion of this possibility is needed. The same issue comes up in Section 6, first paragraph. By design, the author's approach reduces residuals (that is a given). However, can this really be taken as objective evidence that flux or stock predictions have actually been improved?

Other Issues:

1) Why are there so few rain gauge stations applied to the precipitation analysis in Figure 5? It's difficult to believe that only 7 rain gauges are available in the Mediterranean Basin. Also, why is it that the best results (for INT) in Figure 5 occur OUTSIDE of the basin (where INT results are based on an approximate extrapolation)? This seems odd. If INT is performing an accurate downscaling, it seems like it would be more effective WITHIN the basin.

2) In Section 3.2 (on the "optimal selection" (OS) approach) necessary? The methodology section is already quite long and the OS results do not seem to make a significant contribution to the manuscript's results.

3) What is meant by "quasi-triangular balance" in Section 5.1? This terminology will likely be unfamiliar for some HESS readers (it was to me).

4) Figures 4 and 6. A better use of color would be to differentiate between the INT and CAL cases (which are very difficult to distinguish). Also, the INT+/- and CAL+/- notation should be explained in the figure caption.

5) Some discussion of the statistical significant of differences in Figure 3 would be useful.

6) Overall the paper is quite well-written but it does suffer from an excess of minor English usage errors. Superficial proof-reading in this regard would help.

---

## Author Response (AR1)

**Integrating multiple satellite observations into a coherent dataset to monitor the full water cycle Application to the Mediterranean region**

**Editor**

**COMMENTS**

• **I am pleased to say that both reviewers agree on the fact that this paper presents a significant contribution to the scientific community, in particular with respect to the estimation of water balance from satellite observations, that deserves to be published in HESS. However, the reviewers highlighted several majors points that will need to be corrected before this manuscript can be published. Hence, I encourage you to submit a new version of the manuscript, which takes into account the comments made. The new version of the manuscript will be submitted to the reviewers.**
- Thank you for your decision, we hope that the new version of the manuscript is now in a better shape.

• **In addition, I indicate below a few remarks from my side, which generally overlap with the reviewers' remarks.**

• **The paper is sometimes difficult to read due to the description of a large amount of technical information that mask the main message of your work. It is important to describe the scientific objective well throughout the manuscript. For example, the current introduction is very descriptive, lacking the authors' analysis, as well as the presentation of the paper's precise objectives, before the presentation of the plan. Similarly, it is important for each section to recall the scientific context.**
- thank you for this remark, Several updates have been done to underline the main message of our work through the manuscript.

1. the abstract clearly states the article objectives : *The Mediterranean region is one of the more complex environments and is a hot-spot for climate change. The HyMeX (Hydrometeorological Mediterranean eXperiment) aims at improving our understanding of the water cycle at the meteorological to the inter-annual scales. However, monitoring this water cycle with Earth Observations (EO) is still a true challenge: EO products are multiple, and their use still suffer from large uncertainties and incoherencies among the products. Over the Mediterranean region, these difficulties are exacerbated by the coastal/mountainous regions and the small size of the hydrological basins. Therefore, merging/integration techniques have been developed to solve these issues. We introduce here an improved methodology that closes not only the terrestrial but also the atmospheric and ocean budgets. The new scheme allows to impose a spatial and temporal multi-scaling budget closure constraint. A new approach is also proposed to downscale the results from the basin to the pixel scales. The provided Mediterranean WC budget is for the first time based mostly on observations such as GRACE water storage or the netflow at the Gibraltar strait. The integrated dataset is in better agreement with in situ measurements, and we are now able to estimate the Bosporus strait annual mean netflow.*

2. The introduction has been re-written to better focus on methodological aspect with an enhance on the scientific objective and does not present the water cycle equation (Eq. (1)) anymore (The Eq. (1) is now at the beginning of Section 3).

3. The scientific objectives are now propagated along the article and Section 3 is better introduced : *This section presents the integration techniques used to optimize the EO datasets.* As well as Section 4 : *In this section, the obtained integrated datasets are first evaluated in terms of WC budget closure. Our EO datasets integration technique is based on the closure of the WC budget. This is a physical constraint but in some cases (e.g. missing important water component), this constraint could result in a degraded estimation of the components. Therefore, available in situ data (precipitation, evapotranspiration and sea water level) are used to validate some of the water components of the integrated dataset. This evaluation is performed at two different spatial scales: the sub-basin scale and the pixel scale.*

• **In your answers, you indicate that this dataset would be available from the authors. It seems to me that HyMeX has developed**

a strategy to make the data produced available to the scientific community, with different levels of access as appropriate. Even if you do not wish to widely publish this dataset, the use of this database could contribute to better promote your work. Is it possible to use this architecture to make this data available?

- It is agreed that all datasets that are outputs of the WACMOS-MED project will be available on the hymex dataserver. This is now explicit in the conclusion: *The multiple-components dataset INT shows promising aspect for forcing, calibrating or constraining regional models with a water conservation constraint (as required by the community). Some developments and evaluation need are still required before the production of a Climate Data Record (Su et al., 2018) can be started. The two databases (INT and CAL) can however be obtained under request to the corresponding author or via the HyMeX data-server architecture (http://mistrals.sedoo.fr/HyMeX/)*

• **Please check again the Figures and the legends. Some still need to be improved as suggested by the reviewers.**

- Figures and legends have been corrected and color change made.

**Reviewer 1**

**COMMENTS**

• **The MS presents a substantial effort in integrating multiple satellite observations into a coherent data set for monitoring the water cycle of the Mediterranean basin. From a technical point of view, many data products and acronyms are introduced but the reader is rather overwhelmed by the details and misses the central message the MS is trying to convey**

- Thank you for your valuable comments, we tried to simplify a bit the presentation but the introduction of complex notations is necessary. Several updates have been done to make the manuscript simpler:

1. The abstract now focuses on the integration methodologies, which represent the central message of the manuscript : *The Mediterranean region is one of the more complex environments and is a hot-spot for*

*climate change. The HyMeX (Hydrometeorological Mediterranean eXperiment) aims at improving our understanding of the water cycle at the meteorological to the inter-annual scales. However, monitoring this water cycle with Earth Observations (EO) is still a true challenge: EO products are multiple, and their use still suffer from large uncertainties and incoherencies among the products. Over the Mediterranean region, these difficulties are exacerbated by the coastal/mountainous regions and the small size of the hydrological basins. Therefore, merging/integration techniques have been developed to solve these issues. We introduce here an improved methodology that closes not only the terrestrial but also the atmospheric and ocean budgets. The new scheme allows to impose a spatial and temporal multi-scaling budget closure constraint. A new approach is also proposed to downscale the results from the basin to the pixel scales. The provided Mediterranean WC budget is for the first time based mostly on observations such as GRACE water storage or the netflow at the Gibraltar strait. The integrated dataset is in better agreement with in situ measurements, and we are now able to estimate the Bosporus strait annual mean netflow.*

2. The Optimal Selection (OS) Section 3.2 is now suppressed, this method is simply described at the beginning of the SW section 3.3: *A general approach to deal with EO datasets in the analysis of the WC is to choose the best individual dataset for each one of the water components. This is the approach developed, for example, in the GEWEX project. In (Pellet et al. 2017), an Optimal Selection (OS) was based on the minimization of the water budget residuals to select the best combination of individual dataset. Using the OS principle facilitates finding datasets coherent to each other and with independent errors (Rodell et al. 2015). But this kind of strategy limits the use of several source of information to reduce the uncertainties. On the other hand, SW approach benefits from the multiplicity of the observations.*

We hope that the new version of the manuscript is now easier to understand.

**• Since the described project and the MS aims to produce a satellite observations of water cycle, I would suggest that the logic and methods proposed for generating climate data records be followed and organized as such (see e.g. Su et al.,2018, BAMS).**
**-** At this stage, the authors do not pretend yet to produce a Climate Data Record (CDR). The generation of a CDR as presented in Su et al.,2018

(BAMS) is complex, and raises many issues related to long time records (such as absolute or inter-calibration, evaluation procedures, etc.). In our manuscript, we would like to present several EO dataset merging methodologies, discuss their pros and cons. The fact that our dataset is available to the community is for research purpose, and we will consider the full CDR task only after more evaluation, and when we get enough funding to implement such a framework. The abstract is now clearer and does not propose the database anymore, only the conclusion does. Nevertheless, the production of CDR based on a the constrain of the water cycle might be a perspective of our work and this is now explicit in the conclusion: *This multiple-components dataset shows promising aspect for forcing, calibrating or constraining regional models with a water conservation constraint (as required by the community). Some developments and evaluation need are still required before the production of a Climate Data Record (Su et al.,2018, BAMS) can be started. The two databases (INT and CAL) can however be obtained under request to the corresponding author or via the HyMeX data server (http://mistrals.sedoo.fr/HyMeX/).*

• **There is a need to make sure that the used datasets are independent of each other. For example, the GLEAM v3c evaporation dataset is used, but the GLEAM dataset uses also precipitation dataset as input. Could the authors check and verify the independence of such datasets?**
**-** Thank you for this remark. Indeed, the version of GLEAM used in our work is the v3.b (1980-2014) that used multi-source precipitation inputs (TMPA 3B42.v7, MSWEP and ERA-I). The independence of the EO datasets used in an analysis is theoretically desirable, but in practice, this is always difficult to obtain. For instance, most water cycle analyses use a reanalysis (such as ERA-Interim) that does not assure independency between precipitation and evapotranspiration. The merging methodologies that we present are based on the idea that multiple observations should reduce the uncertainties in the estimation of a water component. This is for instance the strategy that is used by ensemble climate models (even if these models are not independent to each other since they use similar physical parameterisations or forcings). It is not a perfect solution but it has advantages. Using the Optimal selection principle could facilitate finding more independent datasets (like in the NEWS project) but we would not benefit from this multiplicity of information. This is now clearer in the text : *Using the OS principle facilitates finding datasets coherent to each other and with independent errors (Rodell et al. 2015). But this kind of strategy limits the use of several source of*

*information to reduce the uncertainties. On the other hand, SW approach benefits from the multiplicity of the observations.*

• **In the first ESA WACMOS project (Su et al., 2014, JAG), an independent evaporation product was generated and is updated continuously. The monthly evapotranspiration for global land area from satellite data (global land 5 km spatial resolution monthly ET dataset, 2000-2017) is produced with a revised SEBS algorithm (Su et al., 2002, HESS; Chen et al. 2013, JAMC) with input as MODIS LST, NDVI, Global forest height, GlobAlbedo and meteorology from ERA-I. A recent comparison of the SEBS ET has reported by Bhattarai et al, 2018, HESS, with the MOD16 ET and a method by integrating radiometric surface temperature (TR) into the Penman-Monteith (PM) equation (STIC). The authors are advised to take a look. The data may be accessed at: http://en.tpedatabase.cn/portal/MetaDataInfo.jsp?MetaDataId=249454.**
**-** Thank you. The SEBS evapotranspiration estimate (Su et al., 2002) presents the major advantage of not computing the relative evaporation based on a surface index, precipitation is not used as an input. This estimation is different to others evapotranspiration estimates based on PM and PT equation. The use of such dataset is a nice perspective that is now mentioned in the conclusion section in the context of closing the water cycle within the energy cycle: *There are still large uncertainties on the water cycle components but the INT methodology appear to be a valuable approach, in particular to include coherency among these components. Several improvements will be considered in the near future: (1) more accurate in situ observations (e.g. Bosporus netflow estimate or coastal discharges) should lead to improved estimates. (2) New water cycle inputs could be considered (e.g. ground water exchange or horizontal exchange at oceanic sub-basin scale) to better characterize the flux and stock terms in the WC. (3) The use of other source of EO estimate would be considered. For example, the evapotranspiration estimate based on the closure of the energy cycle (SEBS algorithm, Su et al., 2002, HESS; Chen et al. 2013, JAMC)) could be tested. This dataset could be a opportunity to (4) close simultaneously the water and the energy cycles and should lead to a better estimate of the evapotranspiration over land.*

• **The authors presented statistics as a quality criteria of the WC closure. I suggest spending some effort in checking the dynamics and physics of the different datasets. I am not sure the correlation coefficients and RMSDs are the most suitable relevant statistical**

**criteria for spatial-temporal datasets.**
- The correlation coefficient and RMSD are classic quality criteria. For example, Pan and Wood (2012), Sahoo et al. (2011), and Zhang et al. (2016) use correlation and RMSD to compare spatial dataset. We also considered the $R^2$ and Mean Absolute Error (not shown) but this was adding no information, just adding confusion. To evaluate the dynamics of the products, we used in Section 4.2 the EOBS precipitation dataset. Beyond the evaluation of the coherency in the water cycle closure (Section 4.1) at monthly scale, the coarse temporal resolution limits deeper evaluation of the dynamic such as extreme rainfall which can hardly be analyzed at monthly scale.

• **The English needs improvement. There are lots of typos and awkward expressions.**
- The typos and English writing have been improved, we hope that the manuscript is now in a better shape.

**Reviewer 2**

**MAJOR COMMENTS**

• **Overall, I found this to be a strong and well-written paper. It makes two (worthwhile) contributions. First, a methodological contribution regarding the appropriate approach for simultaneously enforcing water closure within land, atmosphere and ocean domains. Second, it advances the state-of-the-art in terms of water balance estimates for the Mediterranean Basin. These contributions are significant and worthy of publication in HESS. Nevertheless, there are three major points that the authors should address before final publication. I suspect that some of my concerns arise from misunderstandings on my part (rather than actual flaws) and can be addressed via re-writing for improved clarity.**
- Thank you for your valuable comments, we hope that the new version of the manuscript is now in a better shape.

**The paper needs to do a better job of describing the INTegration (INT) methodology and its impact on subsequent stock and flux**

predictions. My understanding is that the INT approach is applied with two aims: 1) to downscale sub-basin scale results down to the pixel scale and 2) to extrapolate (balance-constrained) results OUTSIDE of the Mediterranean. This raises two questions:

• **If INT down-scales and extrapolates outside of the Mediterranean Basin why does it have any impact on sea-level results in Figure 3 (which presumably reflect spatially averaged conditions within the Basin...neither of which are impacted by INT)? To me, it seems like SW+PF and INT should yield the same results for a sea-level metric. However, these results are used specifically to motivate the added value of INT (line 15 of page 17). Is the improvement in INT versus SW attributable to INT? Or would it also occur for SW+PF?**

**-** You are right, the text is not clear enough in this subsection. The altimeter measures are available only for the Mediterranean Sea, so when the closure constraint is applied over the two seas at once, only the Mediterranean Sea (not the black sea) level is used for the evaluation. You are right, INT and SW+PF would give the same sea level estimates if the Black sea was considered within the Mediterranean Sea in Figure 3. There is no inter/extrapolation of the closure for the Mediterranean Sea and the improvement in INT versus SW is attributable to SW+PF, however the representation of the closure impact of the two seas in the Mediterranean Sea is attributable to INT since SW+PF represents the spatial average over the Mediterranean within the Black Sea. We added additional comments in the associated section: *No inter/extrapolation have been used in INT for the "Mediterranean Sea plus the Black Sea" sub-basin and the improvement of INT versus SW is due only to the impact of the closure constraint. Nevertheless, the SW+PF approach closes the water cycle over the Mediterranean within the Black Sea (no information about the Bosporus netflow) and the spatial downscaling in INT is needed to discriminate the closure correction above the two seas.*

• **What exactly is the rational for the extrapolation portion of INT? Why would you ever want to extrapolate?**

**-** This has been a long discussion between the co-authors. The rational is twofold: (1) The extrapolation of a closure constraint is interesting at the technical level because for other regions, or when working at the global scale, some form of inter/extrapolation is required (See for instance: Munier and Aires, A new global method of satellite dataset merging and quality

characterization constrained by the terrestrial water cycle budget, RSE 205, 119-130, 2018). (2) The extrapolation outside of the Mediterranean region also allow us to use more in situ observation for the evaluation, and this helps better testing the generalisation ability of our extrapolation scheme. Another minor justification is that users often prefer to use a simpler dataset with a rectangular domain, especially in the modelling community.

The justification of this interpolation is based on the assumption that most of the water cycle imbalance is due to satellite errors (this assumption is used for the CAL methodology too). The closure constrain is supposed to improve the satellite estimate by reducing the bias and random errors. If no other information is used (such as surface type, see Munier and Aires 2018), the EO errors should have a spatial continuity and it then makes sense to extrapolate results based on this spatial continuity. We added this discussions in section 3.5: *The extrapolation of a closure constraint is interesting at the technical level because for other regions, or when working at the global scale, some form of inter/extrapolation between the monitored sub-basins is required (Munier and Aires, 2018). The extrapolation outside of the Mediterranean region will also allow us to use more in situ observation for the evaluation, this will help better testing the generalisation ability of our extrapolation scheme. The justification of this inter/extrapolation is based on the assumption that most of the water cycle imbalance is due to satellite errors (this assumption is used for the CAL methodology too). The closure constrain is supposed to improve the satellite estimate by reducing the bias and random errors. If no other information is used (such as surface type, see (Munier and Aires 2018)), the EO errors should have a spatial continuity and it then makes sense to extrapolate results based on this spatial continuity.*

**• Why not just apply terrestrial closure (at a minimum) to Northern Europe SW results?**
**-** You are right, Northern Europe is better monitored and river discharge could had been used to constrain Northern Europe basins at the SW+PF stage. As mentioned earlier, we prefer here to perform the main analysis over the Mediterranean basin and then test the extrapolation scheme over well monitored locations. This is now clearer in the text

**• Also, how does this extrapolation contribute to the (bottom-line) analysis in Figure 6?**
**-** The extrapolation does not contribute to the analysis in Figure 6 since only the Mediterranean basins are considered for computing the annual fluxes. This is now clearer in the text, Section 5.1: *The water cycle is analyzed over*

*its natural sub-basin's boundaries.*

• **I presume it facilitates the application of a larger atmospheric water balance analysis, but given that this is a Mediterranean Basin analysis - it seems strange to extrapolate BEYOND the Mediterranean Basin. It would improve the manuscript if this extrapolation step was better motivated. -** We understand your concerns. We hope that our motivation in the extrapolation is now better explained.

**I feel like the paper could do a better job describing its approach to error estimation (and the effect of these estimates on its merging results).**
- There are two error estimations in our paper: *a priori* EO uncertainty assumption, before the merging, and the *a posteriori* uncertainties estimated after the merging. we hope we will not be confused in the following.

• **I was confused by the treatment of EO uncertainty throughout the manuscript. First, in Line 8 of page 8, the manuscript says states "...we considered the same uncertainty for all data sets of given parameters following de-biasing..." However, later in Section 3.3, it seems as if a different uncertainty is assigned to various precipitation estimates when applying Equation (5). Can these descriptions be made more consistent?**
- Sorry for this ambiguity, Eq. (5) gives the general formula, with different uncertainties, but we considered, you are right, same uncertainties in this application over the Mediterranean basin. This is now clearer in section 3.3: *Since no specific uncertainty estimates were available in the literature for the Mediterranean basin, the uncertainties are assumed to have same standard deviation $\sigma_i$ in the following.*

• **in Section 3.4, the authors invoke a filter-based closure constraint that varies as a function of a $\Sigma$ matrix but do not describe how this matrix was derived. If would be helpful if this was clarified.**
- In our approach, we decided to close the water budget with a relaxation term: we assume an uncertainty in the closure. Such a relaxation on a constrain is commonly used in optimization, it generally follows a Gaussian distribution centred with a standard deviation $\Sigma$ chosen *a priori* in a had hoc way. The matrix $\Sigma$ must include the uncertainty for the continental, oceanic and atmospheric water cycles, it is not provided explicitly in Eq. (2):

$$\Sigma = \begin{pmatrix} \sigma_l & \mathbf{0} \\ \mathbf{0} & \sigma_o \end{pmatrix}$$

where $\sigma_l = \begin{pmatrix} 2 & 0 \\ 0 & 2 \end{pmatrix}$ represents the standard deviation of the constrained terrestrial and atmospheric water budget residual over land; and $\sigma_o = \begin{pmatrix} 2 & 0 \\ 0 & 2 \end{pmatrix}$

represents the standard deviations of the constrained oceanic and atmospheric water budget residual over sea. $\Sigma$ assumes no correlation in the imbalance of the 3 water cycles at monthly and annual scales, at sub-basin or entire basin scales.

• **Equation (5) appears to use the temporal standard deviation of individual products (precipitation products in the example given) after seasonal bias correction as a proxy for the magnitude of their random error. This seems like a dangerous assumption. Assume, for example, that you had a precipitation product that simply mimicked the TMPA seasonal climatology (used here as the de-biasing reference). Given large interannual variability in rainfall, this product would be a poor rainfall product to use in a water balance context. However, it would have a low temporal standard deviation, and (therefore) be heavily weighted by Equation (5).**
- Sorry for the ambiguity. Eq. (5) represents the uncertainty of the EO products, not the temporal standard deviation. This is now clearer in the text: *"let us consider the p precipitation observations $P_i$ associated with Gaussian errors $\epsilon_i \sim \mathcal{N}(O, \sigma_i)$ "*

• **Also - on a related point - after you de-bias the precipitation products, does it really matter (for a long-term water balance study) how - or even IF - you merge the products? After de-biasing, they all have the same long-terms means and will thus produce the same long-term water balance analysis.**
- You are right, the seasonal de-biasing is an important step, especially for the precipitation. Although the season de-biased products will have same season, their inter annual, long-term or short term variations will not be the same. This is now clearer in the text: *After the seasonal de-biasing, all the precipitation products will have a similar season, but their long-term trend, inter-annual or monthly variations will still be different. In particular, the*

*seasonal de-biasing will not change the trend of the EO products.*

• **Page 16, Line 3. The PF approach is designed explicitly to reduce closure residuals. So, it is questionable to use the reduction of closure residuals as evidence that that PF approach is "working" or that a closure constraint is necessary. Another possibility is that not all flux and stock terms are being accounted for him. That is, the flux/stock estimates utilized here are actually accurate but nevertheless should not close. Some discussion of this possibility is needed. The same issue comes up in Section 6, first paragraph. By design, the author's approach reduces residuals (that is a given). However, can this really be taken as objective evidence that flux or stock predictions have actually been improved?**

**-** Yes, closure could happen for the wrong reasons, and we could correct fine EO products for compensating for missing components. The assumption is here that the missing components are random and that the merging will reduce their impact. The only way to make the closure constraint is beneficial is to evaluate the process using independent *in situ* data. This is done in our paper for precipitation and evapotranspiration, it was done also in: Combining data sets of satellite-retrieved products for basin-scale water balance study: 2. Evaluation on the Mississippi Basin and closure correction model, Munier, Aires, Schlaffer, Prigent, Papa, Maisongrande, and Pan, JGR Atmospheres, 2014. What we actually test in section 4.1 is that our methodology is doing what it was designed to do, close the water budget. This is now clearer in the text: *As a closure enforcing, the constraint approaches could yield to a closure of the water cycle, but in degrading fine EO estimate for compensating the imbalance. In order to evaluate the performance of the methodologies in improving the EO estimate, the constrained products will be compared with in situ measurements. The following Section is for assessing that the methodologies do close the water cycle as they suppose to do. The impact of hydrological constraint (PF) as well as the INTegration (INT) and CALibration (CAL) processes on the spatial averaging of the water component estimates and the WC budget residuals, over the several Mediterranean sub-basins, is summarized on Figure A.1 in the Apendix.*

• **Why are there so few rain gauge stations applied to the precipitation analysis in Figure 5? It's difficult to believe that only 7 rain gauges are available in the Mediterranean basin.**

**-** The rain gauge stations came from the FLUXNET database. In this way, precipitation and evapotranspiration evaluation are performed in the same

network (36 gauges for precipitation). However, precipitation is also evaluate using the *in situ* gridded dataset Eobs at the basin scale (Section 4.2).

• **Also, why is it that the best results (for INT) in Figure 5 occur OUTSIDE of the basin (where INT results are based on an approximate extrapolation)? This seems odd. If INT is performing an accurate downscaling, it seems like it would be more effective WITHIN the basin.**

- The fact that INT can have best results outside of the basin can be explained by the poor performance of the precipitation estimate over particularly complex topography (mountainous rainfall) or coastal pixels with land/sea contamination due to the coarse spatial resolution of satellite estimates. This have been added in the text: *The evaluation of EO estimate at 0.25° spatial resolution using tower sites should be taken with caution. The poor performance of satellite estimate over particularly complex topography (mountainous rainfall) or coastal pixels with land/sea contamination could explain the difference between the INT estimate and the FLUXNET measurement at this particular location.*

• **In Section 3.2 (on the "optimal selection" (OS) approach) necessary? The methodology section is already quite long and the OS results do not seem to make a significant contribution to the manuscript's results.**

- Thank you for this comment. Following your suggestion, we suppressed the OS section, the method is simply explained at the beginning of section 4.4 on SW: *A general approach to deal with EO datasets in the analysis of the WC is to chose the best individual dataset for each one of the water components. This is the approach developed, for example, in the GEWEX project. In Pellet et al. (2018) an Optimal Selection (OS) was based on the minimization of the water budget residuals to select the best combination of individual dataset. On the contrary, the SW approach relies on the merging of several EO datasets for each water component, in order to reduce their uncertainty.*

• **What is meant by "quasi-triangular balance" in Section 5.1? This terminology will likely be unfamiliar for some HESS readers (it was to me).**

- Sorry, the term "quasi-triangular balance" was really clumsy. We just wanted to mention that Mediterranean WC is mainly driven by the European sub-basins and that African coasts are not contributing so much.

We have replaced the title of section 5.1 by a straightforward description: "Analysis of the Mediterranean WC".

• **Figures 4 and 6. A better use of colour would be to differentiate between the INT and CAL cases (which are very difficult to distinguish). Also, the INT+/- and CAL+/- notation should be explained in the figure caption.**
**-** The reviewer might mean Figures 4 and 5. The colours are now changed and the caption explicits the notations.

• **Some discussion of the statistical significant of differences in Figure 3 would be useful**
**-** The correlation difference is statistically significant at the 70%-level based on the T-test. This has been added to the caption of Figure 3.

• **Overall the paper is quite well-written, but it does suffer from an excess of minor English usage errors. Superficial proof-reading in this regard would help.**
**-** The typos and English writing have been improved, we hope that the manuscript is now in a better shape.

[revised manuscript text omitted]